# TRAM: Benchmarking Temporal Reasoning for Large Language Models

## Abstract

Reasoning about time is essential for understanding the nuances of events described in natural language. Previous research on this topic has been limited in scope, characterized by a lack of standardized benchmarks that would allow for consistent evaluations across different studies. In this paper, we introduce TRAM, a temporal reasoning benchmark composed of ten datasets, encompassing various temporal aspects of events such as order, arithmetic, frequency, and duration, designed to facilitate a comprehensive evaluation of the temporal reasoning capabilities of large language models (LLMs). We conduct an extensive evaluation using popular LLMs, such as GPT-4 and Llama2, in both zero-shot and few-shot learning scenarios. Additionally, we employ BERT-based models to establish the baseline evaluations. Our findings indicate that these models still trail human performance in temporal reasoning tasks. It is our aspiration that TRAM will spur further progress in enhancing the temporal reasoning abilities of LLMs.

## 1 Introduction

Temporal reasoning is fundamental for humans to understand the world and distinguish between everyday events. For instance, when given the activities "watching a movie" and "watching a sunset", we intuitively recognize that, though both are time-bound, a movie typically lasts longer than a sunset. Moreover, while movies can be watched repeatedly, sunsets transpire once a day. Such innate comprehension isn't just about sequencing events or understanding durations; it extends to more intricate aspects of time, allowing us to make sense of complex narratives and the causality of events. Despite advancements in natural language processing (NLP) and the advent of large language models (Min et al., 2021; Zhao et al., 2023; Wang et al., 2023), mastering temporal reasoning remains a significant challenge due to its intricate nature, the variability of temporal expressions, and the need for contextual understanding.

Recent works in *temporal reasoning* (TeR) mainly focus on time-sensitive question-answering (Zhou et al., 2019; Chen et al., 2021; Dhingra et al., 2022; Tan et al., 2023). These studies consistently show that, despite significant advancements in NLP, current language models still fall short of human-level performance in this domain. While they highlight various aspects of temporal elements, both explicitly and implicitly, such as order, duration, and time-event relations, many intricate facets of TeR, like understanding temporal narratives and temporal causality, remain less explored. Notably, none of these works have tackled broad aspects of TeR within a unified framework.

To facilitate research in this direction, we present the **T**emporal **R**easoning for large l**A**nguage **M**odel benchmark (or TRAM for short), a collection of ten temporal reasoning tasks. These tasks range from foundational understanding (e.g., duration, frequency) to advanced temporal interpretations and computations (e.g., arithmetic, causality). Each task consists of one or more subtasks, all of which are specifically crafted to assess a model's TeR capabilities across varying levels of understanding and difficulty. In total, our benchmark includes 38 distinct subtasks. TRAM incorporates existing natural language understanding datasets, human-crafted templates and questions, web sources, and program generation, comprising a total of 526.7k questions. Answers have been derived through a combination of expert annotations and programmatic generation. Distinct from previous work on temporal reasoning and in alignment with (Hendrycks et al., 2020), our questions are not designed as generative tasks. Instead, they are formatted as straightforward multiple-choice tests, a format more suitable for evaluating LLMs.

To gain deeper insight into the temporal reasoning challenges posed by TRAM, we extensively evaluated several popular language models. This includes BERT (Kenton & Toutanova, 2019), RoBERTa (Liu et al., 2019), and recent LLMs such as Llama2 (Touvron et al., 2023), PaLM2 (Anil et al., 2023), GPT-3.5, and GPT-4 (OpenAI, 2023). We used limited training data to fine-tune BERT-style models. In contrast, the other models were evaluated through either zero-shot or few-shot standard prompting, as well as chain-of-thought prompting. Our findings show that GPT-4 outperforms in most tasks, reaching an average accuracy of up to 87.8%. However, for certain tasks, there are marked performance disparities among the models. Despite the impressive results of GPT-4, it trails human proficiency by roughly 10%, highlighting significant room for LLMs to further improve their temporal reasoning abilities. Manual error analysis revealed that models particularly struggle with nuanced understanding and interpreting implicit cues across all task categories.

In summary, our contributions are threefold:

(1) We introduce TRAM, a comprehensive collection of ten distinct temporal reasoning tasks presented in a multiple-choice question format. Ranging from foundational temporal concepts to intricate temporal interpretations, TRAM serves as a unified framework to assess the temporal reasoning capabilities of LLMs.

(2) We conduct extensive experiments on TRAM, evaluating leading language models including BERT-style models and contemporary LLMs such as Llama2, PaLM2, GPT-3.5, and GPT-4. Our results reveal that even the most advanced LLM falls short of human-level performance, underscoring the opportunities for continued research in this area.

(3) Through manual error analysis of results from TRAM, we highlight the consistent challenges in temporal reasoning faced by current LLMs. Specifically, nuanced comprehension and decoding of implicit temporal cues remain challenging for even advanced models, emphasizing the need for further research to improve the capabilities of LLMs in understanding and reasoning about time.

## 2 RELATED WORK

Our proposal for a comprehensive temporal reasoning benchmark builds on the evolution of datasets in this domain while addressing the lack of a unified system for evaluation. The modern NLP landscape sets the stage for a nuanced evaluation of both BERT-based and LLM paradigms.

**Temporal Reasoning Benchmarks** In the realm of temporal reasoning, several datasets have emerged to address distinct challenges. Early benchmarks, such as TimeBank (Pustejovsky et al., 2003), were predominantly focused on temporal relations. TempEval-3 (UzZaman et al., 2013) broadened the scope by introducing multiple tasks, which included temporal entity extraction and temporal relation extraction. In recent years, there has been a surge in the development of time-sensitive question-answering datasets like MCTACO (Zhou et al., 2019), Time-sensitive QA (Chen et al., 2021), TEMPLAMA (Dhingra et al., 2022), and TEMPREASON (Tan et al., 2023). However, these datasets often specialize in narrower aspects of temporal reasoning, such as duration, frequency, or event-time relations. In contrast, our benchmark offers a comprehensive scope of temporal reasoning, addressing various levels and dimensions of understanding about time. It aims to provide a more complete representation of TeR challenges than previously available datasets.

**Training Paradigms in LLMs** In NLP research, pretraining language models on vast amounts of diverse texts has become standard practice. Through this process, the models encapsulate a broad spectrum of information across various domains. Traditionally, leveraging this pretrained knowledge for downstream tasks primarily involved fine-tuning on task-specific data. BERT-based models like BERT (Kenton & Toutanova, 2019) and RoBERTa (Liu et al., 2019) are representative examples. These models have been applied to a diverse set of tasks, including disease prediction (Zhao et al., 2021), text classification (Wang et al., 2022b), time series analysis (Wang et al., 2022c), and more. However, the introduction of models like GPT-3 (Brown et al., 2020) marked a significant shift away from heavy reliance on extensive task-specific fine-tuning. Instead, the focus has been shifting towards zero-shot and few-shot learning approaches. In these settings, models such as GPT-3 can adapt to new tasks and achieve competitive performance with only a few training examples (Brown et al., 2020). This transition has spurred the development of advanced prompting techniques aimed at enhancing the understanding and reasoning capabilities of LLMs. Some representative prompt-

ing methods include chain-of-thought prompting (Wei et al., 2022), self-consistency (Wang et al., 2022a), tree-of-thought prompting (Yao et al., 2023), and metacognitive prompting (Wang & Zhao, 2023). These techniques guide LLMs to generalize across tasks, ensuring their versatile deployment across a broad spectrum of NLP challenges. In this work, we establish baseline evaluations by considering both traditional BERT-based models and recent advances in LLMs, specifically including Llama2 (Touvron et al., 2023), PaLM2 (Anil et al., 2023), GPT-3.5, and GPT-4 OpenAI (2023). Through this, we aim to provide a comprehensive understanding of their strengths and limitations in diverse temporal reasoning tasks.

## 3 TASKS AND DATASETS

TRAM encompasses ten temporal reasoning tasks, presented as multiple-choice questions (MCQs) across a range of time-related domains. For clarity, we ensure that each question has only one correct answer. The main purpose of TRAM is to spur further research into the advanced temporal reasoning capabilities of LLMs. Overall, these tasks fall under three distinct groups: (1) *Foundational Temporal Understanding Tasks:* Covering basic temporal comprehension, this group incorporates tasks such as ordering, frequency, duration, and typical time. (2) *Temporal Interpretation and Computation Tasks:* Centered on the interpretative and computational aspects of time, this group includes tasks like ambiguity resolution and arithmetic. (3) *Advanced Temporal and Conceptual Understanding Tasks:* Dedicated to exploring intricate temporal relationships and narratives, this category features tasks like relation, temporal NLI, causality, and storytelling. In this work, certain task names, such as 'relation' and 'causality', can have varied interpretations across different contexts. However, they are specifically emphasized for their temporal aspects in this work. Although we might occasionally omit the term 'temporal' for brevity, readers should note that the tasks are centered on time-related elements.

In TRAM, each task is designed with one or more problem types, ensuring diverse representation across data attributes, complexities, and domains. The benchmark includes 526,668 problems in total. For each dataset, we introduce a few-shot development set, with 5 questions per category, and a separate test set for evaluation. Table 1 provides a detailed overview of these tasks, and more details can be found in Appendix B. The majority of tasks employ accuracy as the evaluation metric due to their straightforward MCQ structure. However, for tasks like 'relation' and 'temporal NLI', which exhibit an imbalanced label distribution inherent to their fixed class structure, both accuracy and the F1 score are utilized, even when they are presented as MCQs.

Table 1: Overview of tasks included in TRAM. The "Data Size" column aggregates totals from both the development and test sets. "$K$-Way MC" signifies a multiple-choice response format with $K$ options. *Amb. Res.* denotes Ambiguity Resolution. *NLI* stands for natural language inference. "Same" indicates the text source is the same as the row above.

| Task | Data Size | # Problem Types | Metrics | Answer Type | Text Sources |
|---|---|---|---|---|---|
| Foundational Temporal Understanding Tasks | | | | | |
| Ordering | 29,462 | 2 | Acc. | 3-Way MC | MCTACO[1], Wikipedia, Misc. |
| Frequency | 4,658 | 6 | Acc. | 3-Way MC | MCTACO[1], SQuAD[2], Misc. |
| Duration | 7,232 | 7 | Acc. | 3-Way MC | Same |
| Typical Time | 13,018 | 4 | Acc. | 3-Way MC | Same |
| Temporal Interpretation and Computation Tasks | | | | | |
| Amb. Res. | 3,649 | 5 | Acc. | 3-Way MC | Misc. |
| Arithmetic | 15,629 | 9 | Acc. | 4-Way MC | Same |
| Advanced Temporal and Conceptual Understanding Tasks | | | | | |
| Relation | 102,462 | 1 | Acc./F1 | 3-Way MC | TempEval-3[3] |
| Temporal NLI | 282,144 | 1 | Acc./F1 | 3-Way MC | MNLI[4], SNLI[5] |
| Causality | 1,200 | 2 | Acc. | 2-Way MC | COPA[6], Misc. |
| Storytelling | 67,214 | 1 | Acc. | 2-Way MC | ROC[7], SCT[8] |

[1] (Zhou et al., 2019), [2] (Rajpurkar et al., 2016), [3] (UzZaman et al., 2013),
[4] (Williams et al., 2018), [5] (Bowman et al., 2015), [6] (Roemmele et al., 2011),
[7] (Mostafazadeh et al., 2016), [8] (Mostafazadeh et al., 2017)

## 3.1 Foundational Temporal Understanding Tasks

This group of tasks is fundamental for assessing a model's proficiency in core temporal concepts. For the tasks below, data from the Multiple Choice TemporAl COmmon-sense (MCTACO) dataset incorporates both validation and test sets, while data from the Stanford Question Answering Dataset (SQuAD) dataset includes both training and validation sets. Unless otherwise mentioned, the options for each dataset are generated through a blend of human curation and algorithmic processes, tailored to each specific task. For instance, in the ordering task, correct answers strictly adhere to the accurate chronological sequence of events, while incorrect choices are formed through random permutations. See Figure 1 for example questions of each task.

| | |
|---|---|
| **Ordering** (Facts) | **Q:** Arrange the following events in chronological order: (1) Brusilov Offensive by Russia. (2) Kamehameha I of the Island of Hawaii defeats the Oahuans at the Battle of Nu'uanu. (3) The Kuomintang, the Chinese nationalist party, is founded. (4) Emperor Claudius dies and is succeeded by his grand nephew Nero. (5) St. Norbert and 29 companions make their solemn vows marking the beginning of the Premonstratensian Order. 
 A. (1), (2), (4), (5), (3) ✗   B. (4), (5), (2), (3), (1) ✔   C. (3), (1), (2), (4), (5) ✗ |
| **Frequency** (Commonsense) | **Q:** It is also a love story , between Ace and Tobio, a trans woman. How often do they break up? 
 A. Once ✔   B. Always ✗   C. Once per week ✗ |
| **Duration** (Analogy Inference) | **Q:** While Yoga Session gave attendees time to plant an entire garden, Jazz Concert was enough to water a few plants, and Board Game Night was merely smelling a flower. Which event was the longest? 
 A. Jazz Concert ✗   B. Board Game Night ✗   C. Yoga Session ✔ |
| **Typical Time** (Comparison) | **Q:** Which event typically happens earlier: morning yoga or farmer starting their day? 
 A. Morning yoga ✗   B. Farmer starting their day ✔   C. Around the same time ✗ |

Figure 1: Example questions from temporal ordering, frequency, duration, and typical time tasks.

**Ordering** The temporal ordering task evaluates a model's ability to understand the sequence in which events occur. This task is divided into two primary problem types. For *commonsense* problems, we mainly source questions from the MCTACO dataset (Zhou et al., 2019), specifically targeting subcategories related to temporal ordering. For each individual question selected from this dataset, we pose questions in the format, "Is {candidate answer} possible?" While MCTACO's expected answers are "true" or "false", we introduce another layer of complexity by also including an "undetermined" option. Additionally, we curate another set of commonsense questions, for which sequences of events are manually structured in a logical manner, followed by programmatic question generation. Concurrently, recognizing the significance of tasks rooted in real-world events, we introduce *facts* problems. These focus on major historical events, spanning from ancient to contemporary times, and are sourced from Wikipedia timelines. Models are posed with challenges such as sequencing: "Arrange the following events in chronological order" and verification queries like, "Is the following sequence of events in the correct chronological order?".

**Frequency** The frequency task assesses a model's ability to understand how often events take place over time and comprises six distinct categories of problems. For the *commonsense* category, we source questions from the MCTACO dataset related to frequency. Each selected category ensures the presence of at least two incorrect options and one correct one. To prevent models from memorizing specific answer orders, we randomize the placement of the correct answers. In the *reading comprehension* category, questions are chosen from the SQuAD dataset (Rajpurkar et al., 2016) based on frequency-oriented keywords like "how often", "how many times", and "how frequent". The *application* and *computation* categories are mainly made up of human-curated templates that test the model's ability to infer time intervals and compute either previous or subsequent occurrences. The *comparison* problems blend real and artificially conceived events, challenging the model's ability to differentiate frequency nuances. Lastly, the *facts* category draws questions from various sources, with Wikipedia being the primary one, centering on queries related to events that are known to happen regularly or periodically in either historical or contemporary settings.

**Duration** The duration task is designed to assess a model's capability to comprehend the length of events or periods of time and encompasses seven distinct categories of problems. The *commonsense* problems are derived from the MCTACO dataset, probing the model's fundamental understanding of event durations grounded in everyday scenarios. The extraction methods mirror those used for the "frequency" task. The *reading comprehension* category sources questions from the SQuAD dataset, selecting those with duration-oriented keywords like "how long", "how many years", and "how

much time". Apart from the aforementioned subtasks, all other categories consist of human-curated templates or problems. The *analogy inference* category assesses the model's ability to discern durations through analogical reasoning. The *computation* category tests mathematical precision, where problems often require arithmetic operations to determine event durations. Comparative analysis is examined in two subtasks: *direct comparison*, which demands straightforward judgments of event durations involving both real and artificial events; and *multi-step comparison*, which challenges the model to infer and integrate information across sequential statements. Lastly, the *facts* category primarily draws from Wikipedia, furnishing questions anchored in well-documented historical or contemporary durations.

**Typical Time** The typical time task is constructed to evaluate a model's understanding of when events or activities typically occur, segmented into four distinct categories. The *commonsense* category draws problems from the MCTACO dataset, exploring the model's innate comprehension of event timings as they manifest in daily scenarios. The extraction method for this subtask is similar to that used for the "frequency" task. The *comparison* category, comprising human-curated statements and problems, delves into relative timing. This category involves determining which of two presented scenarios is more temporally typical or discerning which event customarily precedes the other. The *facts* category, primarily sourced from Wikipedia timelines spanning ancient history to the 21st century, provides the model with specific historical or established events and expects it to identify the precise times or periods associated with them. Lastly, the *reading comprehension* problem sets source questions from the SQuAD dataset. This category filters problems based on keywords like "at what time", "when did", and "in what year", challenging the model to extract specific temporal data from passages.

## 3.2 TEMPORAL INTERPRETATION AND COMPUTATION TASKS

This group of tasks is fundamental in gauging a model's adeptness at deciphering, processing, and computing temporal information. See Figure 2 for example questions of each task.

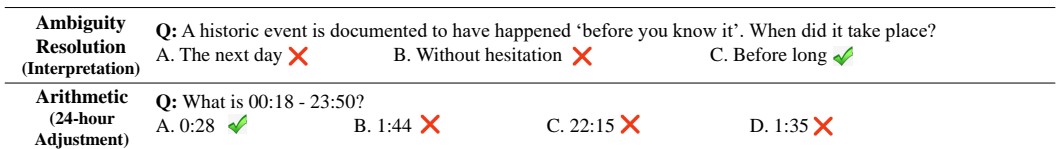

| Ambiguity Resolution (Interpretation) | **Q:** A historic event is documented to have happened 'before you know it'. When did it take place? |
| | A. The next day ✗    B. Without hesitation ✗    C. Before long ✓ |
| Arithmetic (24-hour Adjustment) | **Q:** What is 00:18 - 23:50? |
| | A. 0:28 ✓    B. 1:44 ✗    C. 22:15 ✗    D. 1:35 ✗ |

Figure 2: Example questions from ambiguity resolution and arithmetic tasks.

**Ambiguity Resolution** The temporal ambiguity resolution task aims to gauge a model's ability to decipher and resolve uncertainties related to temporal expressions, divided into five subtasks. The *interpretation* category evaluates the model's comprehension of ambiguous time-related phrases commonly used in everyday language. The *calendar shift* subtask necessitates the conversion between different calendar systems, such as the Julian and Gregorian. The *long-term shift*, *mid-term shift*, and *short-term shift* categories challenge the model's capacity to adjust dates over long (i.e., years), medium (i.e., months, weeks, days), and short (i.e., hours, minutes, seconds) timeframes, respectively. All questions across these categories originate from carefully crafted human templates.

**Arithmetic** The temporal arithmetic task evaluates a model's capacity to manage calculations related to time, organized into nine distinct subtasks. The *application* category presents real-world scenarios such as time calculations involving schooling, vacations, homework, and flights. *Date computation* sets focus on adding or subtracting days from specified dates to determine a new date. *hour adjustment* subtasks, divided into *12-hour* and *24-hour* formats, require the model to calculate time differences or additions. The *month shift* subtask examines the model's ability to pinpoint a month that is a certain number of months away from a specified month. The *week identification* problems determine the exact week number within a year based on a given date. In *year shift*, the model discerns a year a certain number of years relative to a provided year. *time computation* evaluates the model's proficiency in converting various time units, especially over shorter durations like days, hours, minutes, and seconds. Lastly, the *time zone conversion* category requires the model to convert times between different zones. Both the question templates and the programs used to formulate answers derive from human expertise.

## 3.3 ADVANCED TEMPORAL AND CONCEPTUAL UNDERSTANDING TASKS

This group of tasks is fundamental in assessing a model's depth of comprehension in time-oriented narratives and in discerning complex conceptual relationships. See Figure 3 for example questions of each task.

| | |
|---|---|
| **Temporal Relation** | **Q:** Israel wants the EU to arrest any Palestinian suspected of smuggling arms through the crossing, while the EU wants its role to be confined to only monitoring and reporting. What is the relationship between the event 'wants' and the event 'reporting'?
A. ENDED-BY ✗    B. IS_INCLUDED ✓    C. IMMEDIATELY AFTER ✗ |
| **Temporal NLI** | **Q:** Premise: This morning, no doubt, she would have consulted me on the subject, but she had no chance.
  Hypothesis: She would have consulted me on the subject this morning if she'd had the chance.
A. Entailment ✓    B. Neutral ✗    C. Contradiction ✗ |
| **Temporal Causality** (Cause) | **Q:** She noticed that all the wall clocks in the store were set to ten past ten. What's the more plausible CAUSE?
A. It is a common display setting for clocks and watches. ✓  B. It was ten minutes past ten at that moment. ✗ |
| **Temporal Storytelling** | **Q:** I woke up so late this morning. I was panicked when I saw what time it was. I had to be at work on time. I threw myself together quickly. Which of the two endings is the most plausible correct ending to the story?
A. I was able to get a job at a local restaurant. ✗    B. I was still thirty minutes late. ✓ |

Figure 3: Example questions from relation, temporal NLI, causality, and storytelling tasks.

**Relation** The temporal relation task seeks to assess a model's ability to identify the relationship between two entities involving time, categorized as either an *event-to-time* or an *event-to-event* association. Questions are crafted based on the TempEval-3 Silver dataset (UzZaman et al., 2013). The context sentences, which contain the two entities in question, are directly extracted from the original passages. One inherent challenge of this task lies in the subtle nuances among the fixed set of relations. For instance, distinguishing between relations like "BEFORE" and "IMMEDIATELY BEFORE" can be particularly demanding, as they require fine-grained comprehension of temporal sequences. With the predetermined relations from the dataset, the correct relation option is randomized in its placement, while distractor options are chosen from the pool of remaining relations.

**Temporal NLI** The Temporal NLI task is designed to evaluate a model's ability in *natural language inference*, with a particular emphasis on statements that involve temporal elements. We source questions from prevalent NLI datasets, including Stanford Natural Language Inference datasets (SNLI) (Bowman et al., 2015) and Multi-Genre Natural Language Inference (MNLI) (Williams et al., 2018). Data from the MNLI dataset includes training and validation sets, while data from the SNLI dataset includes training, validation, and test sets. We select problems based on keywords that capture a range of temporal nuances, such as explicit references (e.g., 'tomorrow', 'later'), months (e.g., 'May', 'October'), seasons (e.g., 'summer', 'winter'), periods (e.g., 'decade', 'century'), and temporal actions (e.g., 'in advance', 'postpone'). Consistent with the original task, the three response options for all questions are: "Entailment", "Neutral", and "Contradiction".

**Causality** The temporal causality task assesses a model's capability to discern cause-and-effect relationships within scenarios influenced by time. Drawing inspiration from the Choice of Plausible Alternatives (COPA) dataset (Roemmele et al., 2011), we select questions that naturally contain temporal elements such as 'postpone', 'tomorrow', 'summer', and 'clock'. Additionally, we manually craft problems to highlight the temporal nature of COPA-style questions. Each problem presents a situation that revolves around time, followed by a question pinpointing either the most plausible *cause* or *effect* of that situation. Both options for these problems are carefully created by hand. For augmentation purposes, we create additional, mirrored instances for each original sample. This approach ensures that for a given question with two options, each option is supported by a uniquely tailored premise, effectively creating a distinct and relevant context for both choices.

**Storytelling** The *temporal storytelling* task is designed to assess a model's ability to predict the appropriate ending of stories that emphasize temporal elements. We source questions from the ROCStories (ROC) (Mostafazadeh et al., 2016) and Story Cloze Test (SCT) (Mostafazadeh et al., 2017) datasets. We identify and select stories that contain notable temporal components by filtering them using keywords such as 'now', 'tomorrow', 'future', 'always', and 'postpone', among others. The typical format of the task presents a story comprising four sentences, followed by two potential

endings. The model is required to choose the most appropriate conclusion for the story. In the case of SCT, which inherently provides two endings for each story, our focus remains on selecting stories with evident temporal aspects. To further enrich our dataset, we take the initial four sentences from the ROC and employ GPT-2 (Radford et al., 2019) to produce an alternate, incorrect ending, initiated with the prompt "unexpectedly". Subsequently, we filter this augmented data to ensure that stories emphasize the desired temporal themes.

# 4 EXPERIMENTS

In our evaluation, we compare the performance of prevalent LLMs across all datasets and analyze the mistakes they make. We report the best results after multiple runs for each experimental setting.

## 4.1 EXPERIMENTAL SETUP

We evaluate the performance of several well-known language models on the TRAM benchmark, which is organized into two main categories. In the first category, we employ four popular large language models: the open-source Llama-2-13b-chat (Touvron et al., 2023) and the closed-source models PaLM-bison-chat (Anil et al., 2023), GPT-3.5-turbo, and GPT-4 (OpenAI, 2023). Each of these models is accessed using its corresponding API key. Given the constraints of API costs, and following the methodology of (Tan et al., 2023), we assess model performance on 200 examples for each category of each task randomly selected from the test set. For categories with fewer than 200 examples, we utilize all available test examples. For all evaluations, greedy decoding (i.e., temperature = 0) is applied during model response generation. We evaluate each model using two prompting strategies: standard prompting (SP) (Brown et al., 2020; Kojima et al., 2022) and chain-of-thought (CoT)(Wei et al., 2022) prompting. Under both strategies, the models undergo tests in zero-shot and 5-shot settings. In the 5-shot scenario, exemplars are consistently drawn from the development set. Step-by-step answers associated with CoT prompting are obtained through human annotation. More details about prompts can be found in Appendix C.

In the second category, we consider minimal supervision as opposed to traditional fully supervised learning in order to establish baseline evaluations. The rationale behind this decision is driven by the intention to leverage the inherent world knowledge of the models and to ensure an equitable comparison with the previously mentioned LLMs. For this category, we employ four representative BERT-style models, including BERT-base, BERT-large (Kenton & Toutanova, 2019), RoBERTa-base, and RoBERTa-large (Liu et al., 2019). Specifically, for the temporal NLI task, we employ the Sequence Classification variant of BERT and RoBERTa from Huggingface, given its suitability for the task's structure. However, for the other tasks, we utilize the Multiple Choice variant of BERT and RoBERTa from Huggingface. The data sampling strategy for minimal supervision is structured based on the size of the original dataset. For datasets with around 1k samples, we randomly select 50% of the remaining data after setting aside the test data used for LLM evaluation. For datasets with sizes between 3k and 10k, we select 10%. For those with sizes between 10k and 100k, we sample 2.5%, and for datasets with more than 100k examples, we take 1%. This limited training data is then used for the fine-tuning of models. The same test set is used consistently with LLMs.

In addition to evaluating model performance, multiple expert annotators worked on each problem type for every task in TRAM to better understand human performance. Each expert answered a subset of the 50 questions from each category of every task, which were randomly selected from the test set. Collectively, they tackled about 1,900 questions across TRAM. Further details on human expert annotators and human non-specialists are provided in Appendix A.

## 4.2 OVERALL PERFORMANCE COMPARISON

We compared the performance of different models across ten tasks, as shown in Table 2. There are several key takeaways. First, GPT-4 consistently outperforms other models across the majority of tasks, demonstrating a performance advantage of over 15% compared to other models on average. Second, all LLMs show improved performance in the 5-shot setting compared to the zero-shot setting, as expected. Regarding prompting effectiveness, we note that CoT often results in performance enhancements, which corroborates the findings from (Wei et al., 2022), emphasizing the efficacy of step-by-step prompting in augmenting LLMs' performance in intricate reasoning tasks. Third, it is

Table 2: Performance comparison of each model across ten tasks in TRAM. GPT-4 consistently outperforms other models under both zero-shot (0S) and 5-shot (5S) settings across the majority of tasks. Interestingly, the RoBERTa-large model achieves a higher average performance than models with larger architectures, such as Llama2. Human performance serves as an upper bound, illustrating that there still exists room for improvement in LLMs on temporal reasoning tasks. The abbreviations *Freq., Dur., Arith., Rel., Caus.* refer to frequency, duration, arithmetic, relation, and causality, respectively. All values are percentages. Best model results are highlighted in bold.

| Model | Order Acc. | Freq. Acc. | Dur. Acc. | Typical Time Acc. | Amb. Res. Acc. | Arith. Acc. | Rel. Acc./F1 | NLI Acc./F1 | Caus. Acc. | Story Acc. | Average |
|---|---|---|---|---|---|---|---|---|---|---|---|
| Random | 33.3 | 33.3 | 33.3 | 33.3 | 33.3 | 25.0 | 33.3/33.3 | 33.3/33.3 | 50.0 | 50.0 | 35.4 |
| Llama2 (0S, SP) | 50.2 | 71.8 | 63.4 | 72.0 | 45.8 | 51.2 | 34.5/32.3 | 62.7/62.2 | 97.5 | 86.5 | 60.8 |
| Llama2 (0S, CoT) | 51.7 | 73.2 | 64.7 | 73.5 | 48.0 | 54.4 | 39.0/37.7 | 66.0/65.7 | 99.3 | 88.2 | 63.4 |
| Llama2 (5S, SP) | 50.7 | 72.2 | 64.0 | 72.8 | 47.0 | 52.6 | 37.0/35.5 | 63.7/63.2 | 98.8 | 87.3 | 62.1 |
| Llama2 (5S, CoT) | 52.5 | 73.7 | 65.3 | 73.8 | 49.6 | 55.2 | 41.0/39.7 | 66.5/65.7 | 99.5 | 88.5 | 64.3 |
| PaLM2 (0S, SP) | 54.2 | 84.2 | 81.9 | 80.5 | 73.2 | 68.0 | 59.0/58.5 | 68.2/69.1 | 99.3 | 91.2 | 73.9 |
| PaLM2 (0S, CoT) | 55.5 | 85.0 | 82.3 | 81.5 | 74.6 | 69.7 | 62.5/62.1 | 69.3/70.1 | 99.5 | 92.0 | 75.3 |
| PaLM2 (5S, SP) | 55.2 | 84.7 | 82.1 | 81.0 | 74.0 | 68.8 | 61.0/60.7 | 68.5/69.4 | 99.3 | 91.5 | 74.7 |
| PaLM2 (5S, CoT) | 56.2 | 85.2 | 82.7 | 81.8 | 75.0 | 70.2 | 63.5/63.3 | 70.3/71.1 | 99.5 | 92.2 | 75.9 |
| GPT-3.5 (0S, SP) | 52.5 | 77.3 | 71.6 | 78.7 | 72.8 | 72.8 | 40.5/39.1 | 73.8/74.2 | 98.8 | 90.5 | 70.2 |
| GPT-3.5 (0S, CoT) | 53.7 | 78.3 | 72.3 | 79.7 | 74.6 | 74.6 | 44.5/43.5 | 75.2/75.7 | 99.5 | 91.7 | 71.9 |
| GPT-3.5 (5S, SP) | 53.2 | 77.8 | 72.0 | 79.2 | 73.4 | 73.7 | 43.0/41.8 | 74.5/75.0 | 99.3 | 91.0 | 71.2 |
| GPT-3.5 (5S, CoT) | 54.5 | 78.5 | 72.7 | 80.0 | 75.0 | 75.0 | 46.5/45.5 | 75.5/75.9 | 99.5 | 91.7 | 72.5 |
| GPT-4 (0S, SP) | 70.3 | 92.5 | 92.3 | 89.5 | 88.6 | 93.6 | 64.0/63.6 | 89.5/89.8 | 99.0 | 95.8 | 85.7 |
| GPT-4 (0S, CoT) | 71.0 | 93.3 | 92.6 | 90.0 | 89.2 | 93.9 | 67.0/66.6 | 90.5/90.8 | 100.0 | 96.3 | 86.8 |
| GPT-4 (5S, SP) | 70.8 | 92.8 | 92.4 | 89.7 | 89.0 | 93.8 | 66.0/65.6 | 90.0/90.3 | 99.5 | 96.0 | 86.3 |
| GPT-4 (5S, CoT) | **71.5** | **93.7** | **93.0** | **90.2** | **89.8** | **94.3** | 69.5/69.1 | **90.7/91.0** | **100.0** | **96.3** | **87.4** |
| BERT-base | 50.0 | 47.3 | 50.0 | 53.0 | 36.6 | 25.9 | 86.5/86.6 | 53.0/53.4 | 81.0 | 79.0 | 58.5 |
| BERT-large | 52.5 | 53.1 | 53.3 | 56.8 | 37.4 | 28.3 | 89.5/89.5 | 59.5/60.1 | 85.0 | 81.3 | 62.2 |
| RoBERTa-base | 50.8 | 54.5 | 51.8 | 55.3 | 37.4 | 26.4 | 87.0/86.8 | 64.5/64.9 | 82.3 | 81.3 | 61.9 |
| RoBERTa-large | 55.5 | 57.7 | 55.4 | 60.0 | 41.0 | 29.1 | 90.0/90.0 | 70.0/70.3 | 88.0 | 84.0 | 65.9 |
| Human | 86.0 | 96.3 | 97.7 | 94.5 | 94.8 | 98.7 | 96.0/96.0 | 92.0/92.4 | 100.0 | 98.0 | 95.2 |

notable that RoBERTa-large, despite its size, surpasses the larger Llama2 in average performance. This observation underscores that sheer model size doesn't always equate to superior performance. Several factors might contribute to this outcome. RoBERTa-large might utilize optimization strategies that are especially beneficial for these tasks. Additionally, inherent features or efficiencies in its architecture might enhance its ability to understand and process temporal cues. Delving deeper into task-specific performance, certain tasks such as ambiguity resolution and arithmetic show considerable variance across models. For LLMs, performance on the arithmetic task varies significantly, ranging from 51.2% to 94.3%. Moreover, BERT and RoBERTa exhibit exceptional performance in the temporal relation task, potentially due to their bidirectional contextual processing and emphasis on token-level relationships. Their attention mechanisms also allow them to discern and prioritize essential segments in sentences indicative of temporal relationships. This contrasts sharply with their average or below-average performance in other tasks. This discrepancy suggests that some models may be equipped with architectures or training methodologies tailored for certain types of reasoning, or that specific tasks require a distinct understanding not universally handled proficiently by all models. Finally, while GPT-4 leads among all the models, human expertise still exceeds it by roughly 10%, highlighting the complexity of these temporal reasoning tasks and indicating room for future improvements in LLMs.

## 4.3 ERROR ANALYSIS

To better understand the mistakes made by models, we manually analyzed instances where a model, whether in a 0-shot or 5-shot setting or under SP or CoT, made an incorrect choice. We prompted the model to explain its decisions, then reviewed these explanations to identify errors, understand the reasons behind them, and categorize them into specific error types.

For this analysis, we focused solely on LLMs, excluding BERT-style models. Figure 4 showcases the prevalent error types and their respective proportions for each task group. Within the foundational temporal understanding tasks, "assumption bias" was the most frequent error, accounting for 32% of all mistakes. In the interpretation and computation tasks, "calculation slips" dominated, making up 42% of the errors. "Implicit oversights" led in the advanced temporal understanding tasks with a representation of 34%. Detailed descriptions of each error type can be found in Appendix D.

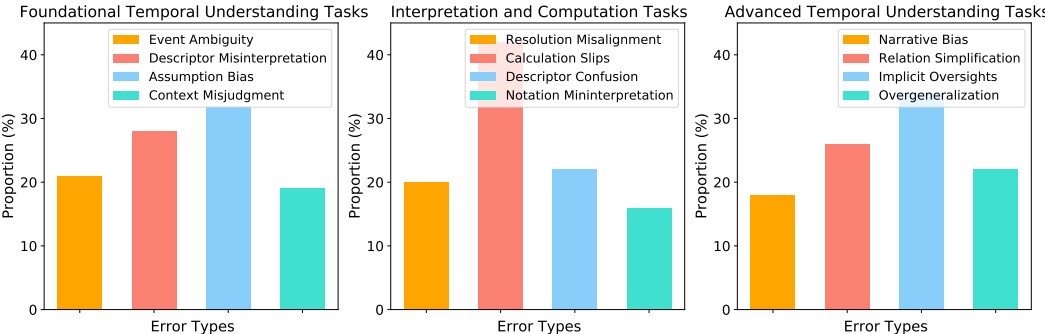

Figure 4: Error type distribution for three groups of tasks in TRAM. Models often struggle with subtle details and hidden clues across all categories.

## 5    DISCUSSION

We introduce TRAM, a comprehensive benchmark spanning ten diverse tasks, to evaluate the temporal reasoning of LLMs. The contrasting performances across models emphasize the significance of experimental strategies and shed light on the intrinsic challenges. This benchmark serves as a tool for researchers to identify model limitations and guide further advancements in this domain.

**Implications of TRAM** The introduction of TRAM establishes a new paradigm for probing the temporal reasoning capabilities of LLMs. Unlike previous benchmarks, which often offered fragmented insights into temporal tasks, TRAM provides a comprehensive system. This allows for a unified evaluation of how models comprehend both rudimentary temporal concepts and complex temporal narratives. The differentiation in task complexity within TRAM elucidates the various stages of temporal understanding. In particular, TRAM underscores challenges like decoding implicit temporal cues and navigating intricate temporal relationships, providing a roadmap for future improvements in LLMs in this area.

**Model Performance and Challenges** Model experimental strategies notably influence large language models' temporal reasoning capabilities. The superior performance in the 5-shot setting, compared to zero-shot, underscores the crucial role of context-specific learning in enhancing these models' grasp on temporal aspects. Moreover, the effectiveness of the CoT prompting highlights the potential of specialized strategies in refining their prowess in complex temporal reasoning tasks.

However, size doesn't inherently guarantee success. The average performance of RoBERTa-large outperforms the larger Llama2, raising intriguing questions about the balance between model size and efficiency. In addition, varied performance across tasks indicates the challenges of crafting a universally adept model for all TeR problems. This variability, combined with the gap between GPT-4 and human expertise, signals ongoing challenges and the need for nuanced improvements.

**Limitations** While TRAM presents a holistic approach to temporal reasoning assessment, we acknowledge its limitations. One primary concern is the subset evaluation of the test set, which may not reflect the full spectrum of LLMs' temporal reasoning capabilities. Furthermore, given the MCQ format, there is a possibility that LLMs could resort to random guessing, rather than genuinely exhibiting temporal reasoning. Such tendencies may mislead the performance evaluation. In addition, textual questions may not capture the entire complexity of temporal reasoning tasks, as real-world scenarios often integrate multi-modal cues such as images and videos.

**Future Directions** TRAM has initiated a step towards evaluating LLMs' temporal reasoning capabilities, but there are further avenues to explore. Going forward, we will experiment with more test data and refine tailored prompting techniques for each task through iterative testing. Moreover, we plan to expand the benchmark to include varied question formats. For generative tasks, this might encompass short answers and summarization. Even within MCQs, we intend to incorporate questions that may have one or more correct answers, allowing for a more comprehensive evaluation. We also plan to fine-tune existing open-source LLMs on these tasks, such as Llama2. These efforts aim to create tailored LLMs that can better understand and reason about time across various contexts.

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

# A  HUMAN ASSESSMENT

In this section, we provide additional details on human participation in our benchmark, including the selection process for experts, verification of their capabilities, and a performance comparison with non-specialists.

**Selection of Expert Annotators** Our selection criteria for expert annotators emphasized a balanced proficiency in both temporal reasoning and quantitative analysis. We included professionals with advanced degrees (M.S. or Ph.D.) in disciplines that offer distinct perspectives on our tasks. This included cognitive science and psychology for qualitative understanding of human temporal cognition, crucial for interpreting more subjective aspects of the tasks. We also involved experts in statistics, mathematics, and computer science to address the quantitative complexities inherent in many of our benchmark tasks. This diverse expertise ensured a comprehensive evaluation of the problems within the TRAM dataset from both qualitative and quantitative angles.

**Expertise Verification Process** To ensure the high caliber of our expert panel, we implemented a robust screening process. This involved a thorough validation of their educational qualifications and a careful review of their professional and research experience, particularly focusing on time perception and quantitative problem-solving. Additionally, we administered a preliminary assessment composed of one random problem from each subtask, totaling 37 problems. The passing criterion for this assessment was set at an average accuracy rate of more than 92%, allowing a maximum of three incorrect responses. This stringent benchmark was established to guarantee the experts' capability in accurately addressing the complex problems in TRAM.

**Comparison with Unspecialized Individuals** In addition to expert assessments, we conducted a comparative analysis with human non-specialists to provide a broader perspective on human performance. These non-specialists, sourced from Amazon Mechanical Turk, consisted of individuals without specialized training in temporal reasoning or related fields. They were tasked with responding to the same set of 1,900 questions as the experts. This group achieved an overall accuracy rate of 63.5% across all tasks. This comparison not only underlines the proficiency of our expert panel but also offers insights into the general human ability to tackle temporal reasoning challenges, providing a baseline for non-expert performance in such tasks.

# B  DATASETS

This section details the dataset construction process, including human-crafted templates for program question generation and the use of temporal keywords to filter questions from existing datasets. We also provide more example questions for each task in TRAM. Additionally, for tasks comprising multiple subtasks, we provide their distribution. Note that the following templates do not represent the full spectrum of templates we used when constructing the datasets.

## B.1  DATA CONSTRUCTION

**Ordering** For our ordering dataset, the *facts* problems were derived from actual events extracted from historical timelines on Wikipedia. Specifically, pages such as `https://en.wikipedia.org/wiki/Timeline_of_the_18th_century` served as our primary data sources. These timelines cover events ranging from ancient history to the 21st century, offering a rich foundation for our dataset. We explored dedicated pages for each available century, ensuring a diverse collection of events across various epochs.

**Frequency** For the frequency task, three main subtasks are generated based on templates: comparison, computation, and applications. Each template contains placeholders, denoted by {}, to represent both events and times. Table 3 outlines some representative templates for each subtask. The construction processes for other subtasks are detailed in the main paper.

**Duration** For the duration task, five main subtasks are generated based on templates: multi-step comparison, analogy inference, computation, direct comparison, and facts. Each template contains placeholders, denoted by {}, to represent both events and times. Table 4 outlines some representative templates for each subtask. The construction processes for other subtasks of the dataset are described in the main paper.

Table 3: Major templates are used for constructing the frequency subtasks: comparison, computation, and applications. The symbols {} serve as placeholders for variable inputs, which can represent both events and times.

| Category | Template |
|---|---|
| Comparison | Compare the frequency of {} and {}. |
| Computation | If {} happens {}, how many times will it occur in {} years? 
 {} appears {}. If it was last seen in {}, when will it next appear? 
 {} appears {}. If it took place in {}, when did it previously occur? |
| Application | If a person's job contract has a renewal every {} years, and they started working in {} and renewed it {} times without gaps, until what year is their current contract valid? 
 A solar eclipse happens at least {} times a year. If the first one in {} is in {}, in which month can we expect the next one? 
 If a plant blooms every {} days and it last bloomed on January 1, on what date will it next bloom? 
 A comet passes Earth every {} years. If its last appearance was in {}, when will it next appear? 
 If a magazine publishes a special edition every {} months and the last one was in January, in which month will the next special edition be? 
 A company holds a general meeting every {} quarters. If the last one was in Q1 of a year, which quarter will the next meeting be? 
 A species of cicada emerges every {} years. If they last emerged in {}, when will they next emerge? 
 If a leap year occurs every 4 years and the last one was in {}, when is the next leap year? 
 A festival is celebrated every {} years. If it was last celebrated in {}, when will it next be celebrated? 
 If a building undergoes maintenance every {} months and the last maintenance was in January, which month will the next maintenance be? |

**Typical Time** For the typical time task, we crafted pairs of time-related events to test the model's proficiency in determining "Which statement is more typical in terms of time?" For instance, when presented with statements such as "People often have dinner in the early to late evening" and "People often have dinner in the mid to late afternoon", the model is prompted to recognize which one is more aligned with a conventional behavior. Similarly, it might evaluate statements like "Bars are typically busiest on Friday and Saturday nights" in comparison to "Bars are typically busiest on Sunday and Monday nights". Through these examples, we aim to assess the model's aptitude in discerning standard temporal practices.

**Ambiguity Resolution** For the ambiguity resolution task, we introduced templates to test the model's proficiency in resolving temporal ambiguities. Additionally, we manually gathered both common and uncommon temporal expressions that might perplex individuals and the model alike, such as "for a coon's age", "when pigs fly", and "in the nick of time". Table 5 presents representative templates for each subtask. Each template contains placeholders, denoted by {}, to represent both events and times.

**Arithmetic** We mainly adopted a programmatic generation approach, grounded in meticulously designed templates that focus on specific temporal calculations. These templates encompass a variety of temporal arithmetic tasks, ranging from basic time adjustments to more complex calculations like week identifications and real-world applications. Table 6 shows the major templates we use for constructing the arithmetic datasets. The variable values, denoted by {}, are randomly generated by programs. Through these templates, we can generate diverse questions that test a model's proficiency in handling different temporal arithmetic scenarios.

**Relation** To derive temporal relation questions from the TempEval-3 Silver dataset, we iterated through each temporal link (*tlink*) to extract the relationship type (*relType*) and relevant event and time IDs. For each *tlink*, the associated *eventInstanceID* provided the *eventID*, either directly or

Table 4: Major templates for constructing the duration subtasks:multi-step comparison, analogy inference, computation, direct comparison, and facts. The symbols {} serve as placeholders for variable inputs, which can represent both events and times.

| Type | Template |
|------|----------|
| Multi-Step Comparison | {} goes on for {}. {} is a third of {}, and {} is as long as {} and {} combined. Which event lasts the longest? 
 Between {} that lasts {}, {} that is four times longer, and {} that's half the total duration of the two, which is the shortest? 
 {} spans {}. {} is double that, but {} is only a third of {}. Which has the most extended duration? 
 If {} lasts {}, {} is twice as long, and {} is half of {}, which event has the medium duration? 
 {} lasts for {}. {} is half of {}'s duration, and {} is triple the combined length of both {} and {}. Which event has the shortest duration? |
| Analogy Inference | During {}, the audience had a chance to enjoy a long opera, while {} showcased just one act, and {} played only an overture. Which event was the shortest? 
 People could indulge in a seven-course meal during {}, a three-course meal in {}, but only an appetizer during {}. Which event was in the middle in terms of duration? 
 {} felt like watching an epic trilogy, {} was more of a feature-length film, and {} was just a brief trailer. Which event was probably the longest? 
 Participants at {} went through an entire yoga session, {} allowed for a short warm-up, while {} was only a few stretches. Which event was the shortest? 
 During {}, attendees could finish a whole board game, in {} they played just a few rounds, and in {} merely set up the pieces. Which event was likely the longest? |
| Computation | The duration of {} is {}. If {} is a quarter shorter than {} and {} is four times the length of {} for {}, how long do all the activities last? 
 For {}, {} takes {}. If {} is twice that duration minus 10% of {}, and {} is half of the sum of {} and {}, how long is the whole event? 
 The total duration of {} is four times the time of {} which is {}. If {} is half of {} minus 5% of {} and {} is twice {} plus 15% of {}, how long do the {} and {} together take? 
 In {}, {} is twice as long as {} which takes {}. If {} is the difference between {} and {}, how long in total? 
 For {}, {} lasts for {}. If {} is double that duration minus 15% of {} and {} is the sum of {} and {} divided by 2, what's the entire duration? |
| Direct Comparison | Which event lasted longer: {} or {}? 
 Which event lasted the longest: {}, {}, or {}? |
| Facts | How long did {} last? |

via the *MAKEINSTANCE* tag. We then identified the sentence containing this event as its contextual background. Using the gathered data, we crafted questions such as "What is the relationship between the event '$event_1$' and the event '$event_2$'?" or analogous questions pertaining to event-time relationships. The context, encompassing both events, was attached to the resulting question to ensure clarity.

**Temporal NLI** To construct our temporal NLI dataset, we adopted a keyword-based filtering approach from SNLI and MNLI datasets. Recognizing that NLI tasks can often hinge on nuanced temporal cues, we curated a comprehensive set of temporal keywords, as shown in Table 7. This selection was designed to capture a broader range of temporal relationships and nuances. Instances

Table 5: Major templates used for constructing the ambiguity resolution dataset. The symbols {} serve as placeholders for variable inputs, which can represent both events and times.

| Type | Template |
|------|----------|
| Short-term Shift | Your plane is supposed to depart at {}. If it's preponed by {}, when is the revised departure? 
 The meal was promised to be on the table at {}. If it's going to be {} postponed, when can you expect to dine? 
 You have an exciting date at {}. If you're lagging by {}, when will you probably meet your date? |
| Mid-term Shift | The match initially set for {} has now been advanced by {}. Which day is it on now? 
 Your usual spa day on {} of every week has been postponed {}. When will it be next week? 
 The weekly town hall usually on {} is delayed by {}. When will it happen? 
 The town carnival usually during the {} week of {} will now be {}. About which date is it now? 
 The music fest during the {} week of {} will be held {}. Around which date will it likely be? 
 The product launch in the {} week of {} has been shifted {}. Around when will it likely be? |
| Long-term Shift | The star, predicted to explode in {}, has its explosion postponed by {} years. When is the new prediction for the explosion? 
 The dynasty which fell in {} had risen to power roughly {} years earlier. When was its establishment? |
| Calendar Shift | If the date is {}/{}/{} in the {}, what is the date in the {}? |
| Interpretation | You receive a memo with the timestamp {}. When should you be prepared? 
 A festival is being organized {}. When would that be? 
 A note suggests meeting {}. When is this suggesting? |

containing at least one term from this extended list were considered to possess temporal elements and were thus included for further analysis.

**Causality** Inspired directly by the style of the COPA dataset, our goal was to capture the intricate weave of cause-and-effect relationships shaped by temporal elements. To this end, we prioritized the inclusion of diverse temporal factors in our dataset, encompassing aspects like seasons, specific times on clocks, special occasions, as well as both long-term and short-term causes and impacts. By meticulously crafting problems with these considerations, we have crafted a rich collection that illuminates the nuanced interplay between time and causality.

**Storytelling** To identify stories with temporal nuances from the ROCStories and SCT datasets, we employed a keyword-based filtering approach. The choice of our keyword set, as shown in Table 8, was shaped by the distinctive nature of the datasets and the contexts they encompass. In ROCStories, for instance, storytelling often employs varied and colloquial temporal expressions, necessitating a specific focus in our keyword selection. Stories containing at least one term from the list were considered to have temporal aspects and were subsequently selected for further processing.

## B.2 EXAMPLE QUESTIONS

For additional examples of various tasks, refer to the following figures: Figure 5 for the ordering task, Figure 6 for the frequency task, Figure 7 for the duration task, Figure 8 for the typical time task, Figure 9 for the ambiguity resolution task, and Figure 10 for the arithmetic task. The advanced temporal understanding group, comprising relation, temporal NLI, causality, and storytelling tasks, which have relatively fewer subtasks, are collectively presented in Figure 11.

Table 6: Major templates used for constructing the arithmetic dataset. The symbols {} serve as placeholders for variable inputs, which are randomly generated by programs.

| Category | Template |
|---|---|
| 24-hour Adjustment | What is {}:{} +/- {}:{}? |
| 12-hour Adjustment | What is {}:{} AM/PM +/- {}:{}? |
| Year Shift | Which year comes {} years after {}? 
 Which year was {} years before {}? |
| Month Shift | Which month comes {} months after {}? 
 Which month was {} months before {}? |
| Date Computation | What will be the time {} years and {} months after month {}? 
 If you add/subtract {} days to the date {}, what will be the new date? 
 If you add/subtract {} months and {} days to the date {}, what will be the new date? 
 If you add/subtract {} weeks and {} days to the date {}, what will be the new date? |
| Week Identification | In which week of year {} does the date {} occur? |
| Time Zone Conversion | If it's {} in the source zone, what's the date and time in target zone? |
| Time Computation | Convert {} days into minutes. 
 Convert {} minutes into hours. 
 Convert {} days into hours. 
 Convert {} seconds into hours. 
 Add {} minutes {} seconds and {} minutes {} seconds. 
 Subtract {} minutes {} seconds from {} hours {} minutes. |
| Application | If a person takes a leave of {} days starting from start_date, on which day may the leave end? 
 If a person was {} years {} month(s) old when he joined school and now he is {} years {} month(s) old, for how long has he been in school? 
 If a person is advised to take medicine every {} minutes, how many times will she take the medicine in a day? 
 If a person starts doing homework at {} and finishes at {} PM, how many hours did he spend on homework? 
 If a flight takes off at {} and the duration of the flight is {} hours, at what time will it land? 
 If a person walks at a speed of {} km/hr and after every km, she takes a rest for {} minutes, how many minutes will it take her to cover {} km? 
 How long will it take to travel a distance of {} kilometers in minutes? |

## B.3 SUBTASK DISTRIBUTIONS

As shown in Table 1, if *Problem Types* count exceeds 1, then we consider it a task involving multiple subtasks. Figure 12 illustrates the distribution of subtasks for each temporal reasoning task. In the case of causality, two problem types are evenly distributed, each accounting for 50%.

## C PROMPTS

We utilize both SP and CoT in our experiments with LLMs. For SP, questions are presented directly without the need for additional steps in the prompt. Consider the following example from the storytelling dataset:

"When I was a boy, my parents used to take my brother and me to the park. We would play, have lunch, and just walk around. One day, when all the picnic benches at the park were occupied, we had one. Two police officers approached and asked if they could join us. Which of the two endings is the most plausible correct ending to the story?

Table 7: Keywords used for filtering SNLI and MNLI datasets that contain temporal aspects.

| Category | Keywords |
| --- | --- |
| Explicit References | today, tomorrow, yesterday, now, soon, later, before, after, day, week, month, year, hour, minute, second, morning, evening, night, noon, midnight, anniversary |
| Days of the Week | Monday, Tuesday, Wednesday, Thursday, Friday, Saturday, Sunday |
| Months | January, February, March, April, May, June, July, August, September, October, November, December |
| Seasons | spring, summer, fall, autumn, winter |
| Periods and Eras | decade, century, millennium, epoch, era |
| General Terms | annual, biannual, quarterly, hourly, daily, weekly, quarter, monthly, fortnight, biweekly, bimonthly, semester, trimester |
| Relative Terms | past, future, current, upcoming, recent, lately, ago, in advance, later, previous, next, moment, time, when, while, duration, period, early, earlier |
| Implicit Temporal Actions | wait, postpone, delay, reschedule, expire, due, schedule, begin, start, end, finish, commence, conclude, last, extend |
| Temporal Transitions and Connectors | until, by the time, as soon as, whenever, since, during, whilst |
| Other Temporal Entities | sunset, sunrise, dusk, dawn, midday, eve, annually, eventually, seldom, often, always, never, sometimes, usually, frequently, occasionally, rarely, just, once, still |

| | |
| --- | --- |
| **Commonsense** | **Q:** Mike started his first business, a bakery. Then Mike launched his online cake delivery service. -True/False? 
 A. Undetermined  B. False  **C. True** |
| **Commonsense** | **Q:** Arrange the following events in chronological order: (1) Sarah spoke her first words. (2) Sarah learned to ride a bicycle. (3) Sarah took her first steps. (4) Sarah started kindergarten. 
 **A. (3), (1), (4), (2)**  B. (1), (3), (4), (2)  C. (4), (3), (2), (1) |
| **Facts** | **Q:** Is the following sequence of events in the correct chronological order? (1) The Periplus of the Erythrean Sea, a Graeco-Roman manuscript is written. It describes an established Indian Ocean Trade route (2) War of the Polish Succession. (3) East India Company starts operations in Bengal to smuggle opium into China. (4) Viking state in Russia founded under Rurik, first at Novgorod, then Kiev. (5) China conquers the Kingdom of Tungning and annexes Taiwan. 
 A. True  **B. False**  C. Undetermined |

Figure 5: Example questions on the temporal ordering task.

(A) They were there to take my brother and me to the police station.
(B) They let us operate the police car lights and siren."

For zero-shot SP, the model is simply prompted with the question: "Given the story 'When I was a boy ... they could join us.' Which of the two endings is the most plausible correct ending to the story? (A) They were... or (B) They let us... The answer (A or B) is: { }." For few-shot SP, exemplar answers (A or B) are provided alongside the questions.

The overall SP procedure across all tasks can be summarized in three steps: (1) *Context Provision (if any):* Provide any necessary background information or context that may aid the model in understanding the scenario presented in the question. (2) *Direct Questioning:* Pose the question directly to

Table 8: Keywords used for filtering ROCStories and SCT datasets that contain temporal aspects.

| Category | Keywords |
|---|---|
| Time References | before, after, recently, now, then, earlier, later, today, tonight, yesterday, tomorrow |
| Temporal Intervals | soon, nowadays, currently, presently, eventually, ultimately, suddenly, immediately, momentarily, previously, formerly |
| Recurring Time Periods | periodically, seasonally, daily, weekly, monthly, annually, biennially |
| Fixed Time Periods | century, decade, millennium, year, minute, hour, day, week, month |
| Parts of the Day | morning, noon, evening, night |
| Duration & Frequency | duration, instant, temporarily, intermittently, frequently, always, never, sometimes, often, rarely, usually |
| Starting Actions | begin/begins/began, start/starts/started, commence/commences/commenced |
| Ending Actions | end/ends/ended, finish/finishes/finished, cease/ceases/ceased, expire/expires/expired, elapse/elapses/elapsed |
| Continuing & Delaying | last/lasts/lasted, continue/continues/continued, resume/resumes/resumed, linger/lingers/lingered, postpone/postpones/postponed, procrastinate/procrastinates/procrastinated |

| | |
|---|---|
| **Facts** | **Q:** How often does ICC Cricket World Cup occur?
**A. Every 4 years**       B. Every 5 years       C. Once a year |
| **Comparison** | **Q:** Compare the frequency of 'Veterans Day' and 'Solar eclipse'.
A.  Veterans Day is more frequent **B. Solar eclipse is more frequent** C. Both events are equally frequent |
| **Computation** | **Q:** If 'Annual invisibility cloak fashion show' happens yearly, how many times will it occur in 100 years?
**A. It will occur 100 times**       B. It will occur 103 times       C. It will occur 99 times |
| **Application** | **Q:** A species of cicada emerges every 22 years. If they last emerged in 1914, when will they next emerge?
**A. 1936**       B. 1939       C. 1934 |
| **Reading Comprehension** | Q: There have been six instances as of 2009 in which the exemption process was initiated. Of these six, one was granted, one was partially granted, one was denied and three were withdrawn. Donald Baur, in The Endangered Species Act: law, policy, and perspectives, concluded," ... the exemption provision is basically a nonfactor in the administration of the ESA. A major reason, of course, is that so few consultations result in jeopardy opinions, and those that do almost always result in the identification of reasonable and prudent alternatives to avoid jeopardy." How many times has the exemption process been used, as of 2009?
**A. Six**       B. Eight       C. Five |

Figure 6: Example questions on the frequency task.

the model without any intermediary steps or additional guidance. (3) *Answer Solicitation:* Request the model to choose and provide the most appropriate answer based on the information given.

In contrast, for CoT, zero-shot learning takes inspiration from (Kojima et al., 2022) by instructing the model to "Answer the question step by step". For few-shot CoT, we manually craft the step-by-step process for 5-shot exemplars in the development set. The procedure to approach this problem is as follows:

| Facts | Q: How long did California Gold Rush last? | | |
|---|---|---|---|
| | A. 3 years | **B. 7 years** | C. 10 years |

| Computation | Q: For a conference, planning lasts for 9 months. If preparation is double that duration minus 15% of planning and keynote is the sum of planning and preparation divided by 2, what's the entire duration? | | |
|---|---|---|---|
| | A. 40.5 months | B. 44.5 months | **C. 38.5 months** |

| Direct Comparison | Q: Which event lasted the longest: World War II, U.S. Woman Suffrage Movement, or British Raj in India? | | |
|---|---|---|---|
| | A. World War II | B. U.S. Woman Suffrage Movement | **C. British Raj in India** |

| Multi-Step Comparison | Q: Art Exhibition has a duration of 2 months. Wine Tasting lasts as long as Art Exhibition and Tech Conference combined, where Tech Conference is triple of Art Exhibition. Which event has the shortest duration? | | |
|---|---|---|---|
| | A. Tech Conference | **B. Art Exhibition** | C. Wine Tasting |

| Commonsense | Q: Lennon accuses his father of leaving him again, and then leaves, after telling his father that he won't live with him anymore . How long does this conversation between Lennon and his father take? | | |
|---|---|---|---|
| | **A. 10 minutes** | B. 10 months | C. 6 weeks |

| Reading Comprehension | Q: In Canada, "college" generally refers to a two-year, non-degree-granting institution, while "university" connotes a four-year, degree-granting institution. Universities may be sub-classified (as in the Macleans rankings) into large research universities with many PhD granting programs and medical schools (for example, McGill University); "comprehensive" universities that have some PhDs but aren't geared toward research (such as Waterloo); and smaller, primarily undergraduate universities (such as St. Francis Xavier). How many years does a degree-granting university in Canada spend teaching students? | | |
|---|---|---|---|
| | **A. Four** | B. Five | C. Six |

Figure 7: Example questions on the duration task.

| Facts | Q: In what year(s) did "The Phoenician alphabet is created" occur? | | |
|---|---|---|---|
| | A. 1006 BCE | B. 1096 BCE | **C. 1050 BCE** |

| Commonsense | Q: Then, he pretended he was his father and pretended that he was driving the tractor. What time did he pretend to drive the tractor? | | |
|---|---|---|---|
| | **A. 1:00 PM** | B. at midnight | C. 1:00 AM |

| Reading Comprehension | Q: In 1978 Aboriginal writer Kevin Gilbert received the National Book Council award for his book Living Black: Blacks Talk to Kevin Gilbert, a collection of Aboriginal people's stories, and in 1998 was awarded (but refused to accept) the Human Rights Award for Literature for Inside Black Australia, a poetry anthology and exhibition of Aboriginal photography. In contrast to previous definitions based solely on the degree of Aboriginal ancestry, in 1990 the Government changed the legal definition of Aboriginal to include any: What year was Gilbert awarded for his efforts? | | |
|---|---|---|---|
| | A. 1960 | **B. 1978** | C. 2017 |

Figure 8: Example questions on the typical time task.

(1) *Read the Story Carefully:* Understand the main theme, setting, and characters introduced in the story. The dominant theme appears to be a nostalgic recollection of a family day out at a park.

(2) *Identify Key Elements from the Story:* The protagonist recalls a childhood memory. The primary setting is a park. The mood is both casual and reminiscent. Despite the park being crowded, they have a picnic spot. Subsequently, two police officers approach the family.

(3) *Evaluate Each Proposed Ending:* For the first ending, a sudden and unexpected twist is introduced that deviates from the story's initial light-hearted narrative. This ending lacks context about why they'd be taken to the police station. The second ending maintains the story's casual and friendly tone, presenting a scenario where the police officers engage positively with the family.

(4) *Comparison of the Two Endings:* Both endings involve the police officers, but the first one introduces a jarring twist without adequate prior context. The second ending aligns more consistently with the story's overarching mood and theme.

(5) *Conclusion:* Given the story's tone, setting, and characters, the second ending appears more plausible and contextually appropriate.

| | | | |
|---|---|---|---|
| **Short-term Shift** | **Q:** Your train's regular schedule is 10:53 AM. However, today it's running 58 minutes behind. When will it depart? | | |
| | **A. 11:51 AM** | B. 11:23 AM | C. 11:38 AM |
| **Mid-term Shift** | **Q:** A marathon was supposed to happen this coming Wednesday, but got shifted three days earlier. When will it occur? | | |
| | A: Thursday | **B. Sunday** | C. Tuesday |
| **Long-term Shift** | **Q:** The dynasty which fell in 1830 had risen to power roughly 90 years earlier. When was its establishment? | | |
| | A: 1742 | B. 1745 | **C. 1740** |
| **Facts** | **Q:** If the date is 9/7/1872 in the Julian, what is the date in the Gregorian? | | |
| | A. 6/6/1871 | **B. 9/19/1872** | C. 5/26/1872 |

Figure 9: Example questions on the ambiguity resolution task.

| | | | | |
|---|---|---|---|---|
| **12-hour Adjustment** | **Q:** What is 08:24 AM - 07:42? | | | |
| | **A. 12:42 AM** | B. 1:56 AM | C. 10:31 PM | D. 11:34 PM |
| **Year Shift** | **Q:** Which year comes 11 years after 1718? | | | |
| | A. 1731 | B. 1707 | C. 1764 | **D. 1729** |
| **Month Shift** | **Q:** Which month comes 2 months after December? | | | |
| | A. June | **B. February** | C. January | D. September |
| **Date Computation** | **Q:** What will be the time 16 years and 8 months after August 1412? | | | |
| | A. June 1430 | B. May 1430 | **C. April 1429** | D. June 1431 |
| **Week Identification** | **Q:** In which week of year 2007 does the date 10-12-2007 occur? | | | |
| | **A. Week 41** | B. Week 28 | C. Week 5 | D. Week 10 |
| **Time Zone Conversion** | **Q:** If it's 12 PM on May 4, 1904 in Asia/Kolkata, what's the date and time in US/Eastern? | | | |
| | A. 6 AM on May 4, 1904 | B. 12 PM on May 4, 1904 | **C. 1 AM on May 4, 1904** | D. 11 AM on May 4, 1904 |
| **Time Computation** | Q: Subtract 1 minute 32 seconds from 1 hour 22 minutes. | | | |
| | A. 77 minutes 25 seconds | B. 90 minutes 38 seconds | C. 70 minutes 18 seconds | **D. 80 minutes 28 seconds** |
| **Application** | Q: If a girl is advised to take medicine every 139 minutes, how many times will she take the medicine in a day? | | | |
| | A. 12 | B. 11 | C. 8 | **D. 10** |

Figure 10: Example questions on the arithmetic task.

After defining the step-by-step procedure, we employ it to steer the model's thought process. This structured methodology better prepares the model to reason through the question and formulate a well-considered answer, thereby providing a distinct advantage over the SP method. We structure our prompt as follows: "Begin by reading the story carefully, ensuring you fully understand its main theme, setting, and the characters. {Immediate analysis}. Subsequently, identify the key elements of the story. {Immediate analysis}. Assess each proposed ending within the context of the narrative. {Immediate analysis}. Compare the two endings, highlighting any thematic or tonal discrepancies. {Immediate analysis}. Conclude by determining which ending appears more plausible, offering a rationale for this selection {Immediate analysis}."

In general, the CoT procedure across all temporal reasoning tasks is as follows: (1) *Understanding Context:* Begin by reading the provided data, statement, or story attentively. Understand the overarching theme, objectives, or the problem's primary ask. (2) *Key Elements Extraction:* Identify and highlight crucial elements, specifics, or characters. This could mean different things for different tasks - key events in a story, terms in a mathematical problem, or clauses in a statement. (3) *Evaluation:* Assess the core objective of the problem in its context. This could be understanding the chronology for ordering, assessing frequency, gauging durations, or even understanding the logical or causal flow in more complex problems. (4) *Analysis and Comparison:* If there are multiple options or scenarios presented, conduct a deep analysis. Compare, contrast, and evaluate based on the preceding steps. (5) *Reasoned Conclusion:* Conclude with a structured answer or resolution to the problem, ensuring that the decision-making process aligns with the evidence or data presented. In

| Temporal Relation | **Q:** It added that the Ministry of Economic Affairs and Finance was assigned to draw up practical procedure for the ceding, while the Ministry of Welfare and Social Security would be responsible for identifying the beneficiaries in two months. What is the relationship between the event 'added' and the event 'ceding'? 
 A. IS_INCLUDED    B. SIMULTANEOUS    **C. AFTER** |
|---|---|
| Temporal NLI | **Q:** Premise: Two guys playing football on a campus green. 
 Hypothesis: They are practicing before the big game tomorrow 
 A. Entailment    **B. Neutral**    C. Contradiction |
| Temporal Causality (Effect) | **Q:** The seasons changed from summer to autumn. What's the more plausible RESULT? 
 A. People evacuated their homes.    **B. Leaves fell from the trees.** |
| Temporal Storytelling | **Q:** There is a huge clock in my living room. I turned the clock back one hour for daylight savings. My wife also turned the clock back one hour for daylight savings. Our 2 kids each turned the clock back one hour for daylight savings. Which of the two endings is the most plausible correct ending to the story? 
 **A. Then we wondered why it got so dark so early.**    B. The kids were not happy. |

Figure 11: Example questions on advanced temporal reasoning tasks, including relation, temporal NLI, causality, and storytelling.

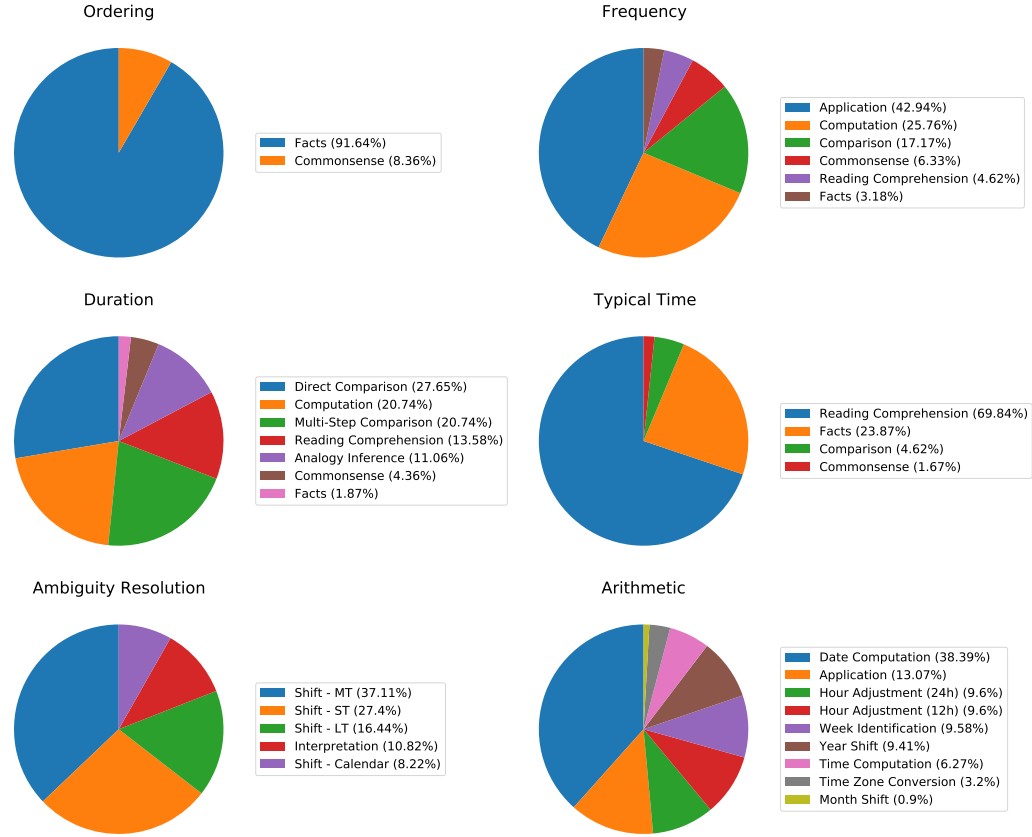

Figure 12: Distribution of subtasks for each distinct temporal reasoning task.

practice, the procedure varies for each task to account for the diverse nature of temporal reasoning tasks.

# D  ERROR TYPES

In this section, we delve into each specific error that LLMs commonly encounter in temporal reasoning tasks, as illustrated in Figure 4.

**Foundational Temporal Understanding Tasks** In foundational temporal understanding, LLMs encounter several distinct challenges. Firstly, *Assumption Bias* is evident when models over-rely on patterns from their training, often neglecting cultural or individual variations. Next, *Temporal Descriptor Misinterpretation* occurs when models misinterpret terms, such as perceiving "often" as a daily event instead of a possible weekly occurrence. *Event Ambiguity* presents another challenge, where events can be described in ways that allow for multiple interpretations, requiring models to select the most suitable one based on context. Lastly, *Contextual Misjudgment* is when models either miss or misinterpret explicit temporal clues, leading to errors in their reasoning.

**Temporal Interpretation and Computation Tasks** In computational and interpretable temporal reasoning, LLMs encounter various challenges. Firstly, *Calculation Slips* highlight instances where models often make calculation mistakes like inappropriate handling of time carries. Following this, *Descriptor Confusion* arises when models misalign qualitative terms such as "seldom" or "frequently" with their quantitative meanings. *Resolution Misalignment* represents the struggle models face with vague time references, such as deciphering the exact duration from terms like "in a while". Lastly, *Temporal Notation Misinterpretation* occurs when models confuse time formats, for example, mixing up AM with PM or not differentiating between 24-hour and 12-hour representations.

**Advanced Temporal and Conceptual Understanding Tasks** In advanced temporal reasoning tasks, LLMs frequently encounter certain pitfalls. Among the most prevalent is *Implicit Oversights*, where models overlook subtle but crucial temporal indications, resulting in inaccurate conclusions. Also, they may face *Relation Simplification*, wherein complex temporal interplays between events are either misunderstood or overly simplified. LLMs might also fall into the trap of *Narrative Bias*, where they overly depend on familiar story patterns, prioritizing recognized sequences over fresh interpretations. Lastly, *Overgeneralization* becomes evident when models incorrectly apply broad temporal conventions to specific situations, leading to misunderstandings when scenarios diverge from the norm.

# E  TASK GLOSSARY DEFINITIONS

In this section, we provide a glossary with definitions of all tasks and subtasks encompassed within our TRAM benchmark for clarity. In our actual dataset formatting, the subcategory (if a task comprises multiple subtasks) or source (if a single subtask is sourced from an existing dataset) is marked for verification and convenient lookup.

**Ordering**: Chronological arrangement of events.

- *Commonsense*: Logical sequencing of events based on general knowledge.
- *Facts*: Accurate ordering of historical events based on factual information.

**Frequency**: Determination of how often events occur over time.

- *Commonsense*: Assessment of event occurrence rates based on general knowledge.
- *Reading Comprehension*: Frequency information extraction from passages.
- *Application*: Inference of time intervals and event frequencies.
- *Computation*: Calculation of event occurrences and intervals.
- *Comparison*: Differentiation of event frequencies in various contexts.
- *Facts*: Identification of periodically occurring events.

**Duration**: Determination of the length of events or time periods.

- *Commonsense*: Evaluation of time spans in everyday life scenarios.
- *Reading Comprehension*: Duration information extraction from passages.
- *Analogy Inference*: Discernment of relative time spans through contextual comparison.
- *Computation*: Calculation of event lengths.
- *Direct Comparison*: Straightforward assessment of event durations in a given set.

- *Multi-step Comparison*: Analysis of relative durations using layered information.
- *Facts*: Identification of length of factual events.

**Typical Time**: Determination of when events or activities typically occur.

- *Commonsense*: Analysis of usual event timings in daily life scenarios.
- *Comparison*: Assessment of relative event timings and typical sequences.
- *Facts*: Identification of historical times or periods from established events.
- *Reading Comprehension*: Specific time information extraction from passages.

**Ambiguity Resolution**: Resolution of uncertainties in temporal expressions.

- *Interpretation*: Understanding of ambiguous time-related phrases.
- *Calendar shift*: Conversion between different calendar systems.
- *Long-term shift*: Adjustment of dates over extended periods (years).
- *Mid-term shift*: Date adjustments over intermediate periods (months, weeks, days).
- *Short-term shift*: Time adjustments over brief periods (hours, minutes, seconds).

**Arithmetic**: Execution of time-related calculations.

- *Application*: Real-world time calculation scenarios (schooling, vacations, etc.).
- *Date Computation*: Addition or subtraction of days to find new dates.
- *12-hour Adjustment*: Time difference calculations in 12-hour format.
- *24-hour Adjustment*: Time difference calculations in 24-hour format.
- *Month Shift*: Identification of a future or past month from a given date.
- *Week Identification*: Determination of week numbers within a year.
- *Year Shift*: Calculation of future or past years from a specified year.
- *Time Computation*: Calculating future or past years from a specified year.
- *Time Zone Conversion*: Conversion of times between different time zones.

**Relation**: Identification of the temporal relationship between two entities, either as an event-to-time or event-to-event association.

**Temporal NLI**: Assessment of a 'hypothesis' as true (entailment), false (contradiction), or undetermined (neutral) relative to a 'premise' with temporal elements.

**Causality**: Analysis of cause-and-effect relationships in time-related scenarios.

- *Cause*: Identification of the initiator or reason leading to a particular event.
- *Effect*: Determination of the outcome or consequence resulting from a specific cause.

**Storytelling**: Prediction of appropriate story endings, with an emphasis on temporal elements.

## F    COMPARISON OF SOURCE VS. CURATED DATASETS

We provide several representative examples sourced from existing datasets, allowing for a comparison between the original sources and our curated datasets. Specifically, Table 9 and Table 10 demonstrate the transformation of original Yes/No binary questions from the MCTACO dataset into our frequency and ordering tasks in MCQ formats, respectively. Meanwhile, Table 11 shows the transformation of original short-answer questions from the SQuAD dataset into our duration task. Our benchmark combines the strengths of existing benchmarks with extensive manual effort, including the addition of distracting or confusing options, the filtering out of irrelevant questions for quality control, and the reformulation of problems, thereby setting a new standard for assessing temporal reasoning in LLMs.

Table 9: Comparison of source (MCTACO) and curated question in TRAM for the Frequency task.

| Source Dataset (MCTACO) | Curated Dataset (TRAM) |
|---|---|
| **Question:** Allan crouched over his desk once more, pen in hand and mind blank. How often does Allan crouch over his desk? | **Question:** Allan crouched over his desk once more, pen in hand and mind blank. How often does Allan crouch over his desk? |
| **Options/Answers:** 

 • Once a second - No 

 • Once two years ago - No 

 • Every day - Yes 

 • Several times per second - No 

 • Daily - Yes | **Options:** 

 • (A) Every day 

 • (B) Several times per second 

 • (C) Once a second 

 **Answer:** 

 • (A) Every day |
| **Commentary:** Binary Yes/No format, simple frequency assessment. | **Commentary:** Transition to an MCQ format enriches the question's complexity by offering closely related alternatives. |

Table 10: Comparison of source (MCTACO) and curated question in TRAM for the Ordering task.

| Source Dataset (MCTAC0) | Curated Dataset (TRAM) |
|---|---|
| **Question:** Church is brought back to life, but is an evil shell of himself. What did Church do next? | **Question:** Church is brought back to life, but is an evil shell of himself. What did Church do next? Is "took a nap" possible? |
| **Options/Answers:** 

 • "took a nap" - No | **Options:** 

 • (A) Undetermined 

 • (B) TRUE 

 • (C) FALSE 

 **Answer:** 

 • (B) Two months |
| **Commentary:** Binary Yes/No format, simple ordering assessment. | **Commentary:** Transition to an MCQ format introduces additional ambiguity and uncertainty into the question. |

Table 11: Comparison of source (SQuAD) and curated question in TRAM for the Duration task.

| Source Dataset (SQuAD) | Curated Dataset (TRAM) |
|---|---|
| **Question:** It was not until January 1518 that friends of Luther translated the 95 Theses ... Within two weeks, copies of the theses had spread throughout Germany; within two months, they had spread throughout Europe. How long did it take for the Theses to spread through Europe? | **Question:** It was not until January 1518 that friends of Luther translated the 95 Theses ... Within two weeks, copies of the theses had spread throughout Germany; within two months, they had spread throughout Europe. How long did it take for the Theses to spread through Europe? |
| **Options/Answers:** 

 • Short answer: Two months | **Options:** 

 • (A) 45 days 

 • (B) Two months 

 • (C) 2 days 

 **Answer:** 

 • (B) Two months |
| **Commentary:** Short-answer format, simple duration assessment. | **Commentary:** Transition to an MCQ format introduces additional numerical ambiguity in problems involving multiple numbers. |

