# OpenReview forum: "TRAM: Benchmarking Temporal Reasoning for Large Language Models"
_ICLR.cc/2024/Conference — Submitted to ICLR 2024_

### Official Review · Reviewer_PXAF · 2023-10-30

**Soundness:** 3 good
**Presentation:** 3 good
**Contribution:** 2 fair
**Rating:** 3
**Confidence:** 4

**Summary:**

This paper focuses on introducing a new benchmark for temporal reasoning tasks in a multi-choice format to evaluate the temporal reasoning capabilities of LLMs. There are 10 tasks collected in the benchmark covering Ordering, Frequency, Duration, Typical Time, Ambiguity Resolution, Arithmetic, Temporal Relation, Temporal NLI, Temporal Causality and Temporal Storytelling. The authors evaluated several leading LLMs including Llama2, PaLM2, GPT-3.5 and GPT-4 against the tasks, and reported the evaluations results. In addition, the authors also conducted error analysis of the results to understand where current LLMs are still struggling in temporal reasoning.

**Strengths:**

The set of tasks is very comprehensive to cover a wide variety of temporal reasoning task types. The dataset size is quite large, bigger than 3K samples for 9 out of 10 tasks, making the evaluation results robust against variance. The LLMs evaluated are also quite comprehensive covering both the SOTA proprietary and the SOTA open-source LLMs. The error analysis is also quite useful for understanding where the current LLMs still fall short in temporal reasoning.

**Weaknesses:**

There are two major weaknesses to this work:
1. The task difficulty of the TRAM benchmark is not enough to serve the purpose for ongoing evaluations of future more powerful LLMs. 8 out of 10 tasks have close to or better than 90% accuracy when evaluated on the SOTA LLM GPT-4 (5S, CoT). For the same 8 tasks, the gap between GPT-4 and Human is within 5%. This leaves very little headroom for improvement to validate future more powerful LLMs.

2. While the benchmark is designed specific for temporal reasoning, it is not entirely clear how much failure in the errors are due to the model's lack of general reasoning capability, rather than specific to the time dimension. A more careful separation and attribution to either general reasoning and temporal reasoning failures is needed to make the benchmark really useful for gauging temporal reasoning progress.

**Questions:**

1. The difference between TRAM and previous temporal reasoning benchmarks is unclear: for instance Duration and Frequency are already covered by earlier benchmarks. A more detailed and convincing comparison is needed to establish the necessity of the newly introduced TRAM benchmark.

2. In Section 4.3, the authors claimed that they prompted the model to explain its decisions, and used these explanations to identify errors, understand the reasons, and categorize error types. This is very concerning as LLMs are known to not know what they don’t know. Relying on LLMs to provide the explanations for error analysis puts a big question mark on the reliability of the error analysis conclusions themselves.

---

> ### Author Response · Authors · 2023-11-21
> **Response to Reviewer PXAF (Part 1)**
>
> Reviewer PXAF, thank you for providing the insightful suggestions. We hope the following responses address your concerns.
>
> **Weakness 1: Difficulty of the benchmark.**
>
> We appreciate your concern about the difficulty level of our TRAM benchmark. The purpose of this benchmark is to provide a comprehensive test that covers a wide range of temporal reasoning abilities for LLMs. As detailed in our paper, most tasks consist of multiple subtasks. Although the average SOTA performance is high, among the eight tasks you mentioned, which total 34 subtasks, there are 12 out of 34 subtasks where the GPT-4 (5S, CoT) setting performs at or below 80%. For instance, in this setting, the Ambiguity Resolution (calendar-shift) task achieves 53% accuracy, the Typical Time (comparison) task 77%, and the Arithmetic (date computation) task 78%. These results underscore specific areas that require further refinement, even in SOTA models.
>
> Moreover, the performance gap between GPT-4 and human benchmarks warrants particular attention. While this gap may seem modest (within 5%), achieving even incremental improvements is notably challenging as models approach human-level accuracy. Therefore, these seemingly small gaps are, in fact, substantial indicators of areas where future LLMs can and need to improve.
>
> Meanwhile, to address your concern and challenge the models more rigorously, we have considered some straightforward and effective methods, including introducing additional options and modifying question formats. To test the efficacy of these methods, we conducted case studies on the Causality and Storytelling tasks, which initially showed high performance. Under GPT-4 (5S, CoT), we reevaluated 50 samples from each task that were previously predicted correctly. When the distracting choices were contextually close to the correct answer, the model's performance dropped, with 9 out of 50 incorrect predictions for Causality and 17 out of 50 for Storytelling. In contrast, when distractors were clearly differentiated from the correct answer, the model had 5 out of 50 incorrect predictions for Causality and 13 out of 50 for Storytelling.
>
> Even in a SOTA setting, our deliberate introduction of additional options can decrease model performance by more than 10%. Additionally, converting some tasks to a short-answer format has been shown to affect performance; for example, the SOTA performance on the Arithmetic task dropped from 94.3% to 89.6%. We are dedicated to continuously refining the tasks in our benchmark and to providing a variety of question formats to meet different levels of model testing. Updates to question sets and results will be regularly shared on our GitHub page.
>
> **Weakness 2: Temporal vs. general reasoning in model errors.**
>
> We designed our temporal reasoning (TeR) benchmark, recognizing the importance in understanding and reasoning about time. Although TeR is a specialized form of general reasoning, our aim is to isolate and evaluate this unique aspect due to its broad use in everyday life. In our error analysis, we distinguish between general reasoning and TeR when evaluating LLM capabilities. For example, in the Frequency task with the question ‘If a person’s job contract has a renewal every 4 years, and they started working in 1857 and renewed it 3 times without gaps, until what year is their current contract valid’, GPT-4’s error was categorized as ‘Contextual Misjudgment’. The model incorrectly assumed that each 4-year contract period concludes at the beginning of the following year, instead of at the end of the fourth year. This error demonstrates a misunderstanding in processing temporal information, rather than a general reasoning flaw. While time-related mistakes may occur within a broader reasoning context, our primary objective is to gauge a model’s progress in TeR.
>
> **Question 1: TRAM vs. existing temporal reasoning benchmarks.**
>
> The following table provides a comparison between our TRAM and previous temporal reasoning datasets, including task coverage and data size. The order of the prior work listed is from the most recent to the earlier ones.

---

> > ### Author Response · Authors · 2023-11-21
> > **Response to Reviewer PXAF (Part 2)**
> >
> > **Question 1: TRAM vs. existing temporal reasoning benchmarks.**
> >
> > | Dataset/Benchmark | Size  | Task Coverage                                                  |
> > |-------------------|-------|---------------------------------------------------------------|
> > | TRAM              | 526.7k| Ordering, Frequency, Duration, Typical Time, Ambiguity Resolution, Arithmetic, Relation, Temporal NLI, Causality, Storytelling |
> > | MenatQA           | 2.9k  | Scope, Order, Counterfactual                                   |
> > | TEMPREASON        | 52.8k | Facts                                                          |
> > | TEMPLAMA          | 50k   | Facts                                                          |
> > | Time-Sensitive QA | 41.2k | Facts                                                          |
> > | TIMEDIAL          | 1.1k  | Commonsense                                                    |
> > | MCTACO            | 13k   | Commonsense (duration, temporal ordering, typical time, frequency, stationarity) |
> >
> > We acknowledge that aspects like Duration and Frequency are indeed covered by previous benchmarks. However, as these are foundational aspects in temporal reasoning (TeR), including these tasks is necessary and essential for a comprehensive TeR benchmark. Compared to earlier works, our dataset offers a much larger data size and broader task coverage, encompassing 10 main tasks and 38 subtasks. For instance, in the classic tasks of ‘Duration’ and ‘Frequency’, earlier works mainly cover Facts and Commonsense. In contrast, our work includes 7 subtasks for ‘Duration’ and 6 subtasks for ‘Frequency’, delving into more nuanced aspects of these temporal elements, which are less emphasized by earlier works. Consequently, the extensive data size of TRAM and the intricate design of its subtasks represent a notable change in both the depth and breadth of temporal reasoning (TeR) evaluation, compared to existing benchmarks.
> >
> > **Question 2: Error analysis procedure.**
> >
> > Thank you for sharing your concern. Regarding your statement that “LLMs are known to not know what they don’t know”, we would like to offer a perspective that frames this phenomenon within the context of ‘hallucination’ in model outputs. Hallucination typically refers to models generating plausible but incorrect or irrelevant information. While this is a known limitation, it does not significantly impede our objective to understand why the model makes incorrect predictions. In fact, analyzing these ‘hallucinations’ can be insightful in pinpointing areas where the model’s reasoning deviates from expected logic or factual accuracy.
> >
> > Furthermore, evolving research in AI around concepts like ‘self-refinement’ [1], ‘self-debugging’ [2], and ‘self-reflection’ [3] suggests that models may possess some rudimentary self-assessment capabilities. Therefore, prompting a model to explain its reasoning aligns with these emerging capabilities and is a useful tool for error analysis. These explanations, when combined with expert review, form a reliable approach to understanding model decisions.
> >
> > Below is an example illustrating how we identify the reason behind a model’s error:
> >
> > This problem falls under the Frequency task: “If a person’s job contract has a renewal every 4 years, and they started working in 1857 and renewed it 3 times without gaps, until what year is their current contract valid? Options: (A) 1873 (B) 1874 (C) 1870”. The correct answer is (A), but GPT-4 initially provided the incorrect answer (C). Upon querying the model for its reasoning, it responded: “I would have assumed that each 4-year contract period concludes at the beginning of the following year, rather than at the end of the fourth year. Starting from 1857, the first period would seemingly end as 1861 begins, the second as 1865 begins, and the third as 1869 begins, leading me to conclude that the contract is valid until the start of 1870”. This response indicates that the model misunderstood the completion year of the contract. We categorize this error as ‘Contextual Misjudgment’, where the model misinterprets the explicit temporal context, leading to a flawed conclusion.
> >
> > Thus, while we acknowledge the limitations of LLMs in terms of self-awareness, we believe that their generated explanations still hold value for error analysis, especially when supplemented with rigorous human-led methodologies.
> >
> > [1] Madaan, Aman, et al. "Self-refine: Iterative refinement with self-feedback." arXiv preprint arXiv:2303.17651 (2023).
> >
> > [2] Chen, Xinyun, et al. "Teaching large language models to self-debug." arXiv preprint arXiv:2304.05128 (2023).
> >
> > [3] Shinn, Noah, Beck Labash, and Ashwin Gopinath. "Reflexion: an autonomous agent with dynamic memory and self-reflection." arXiv preprint arXiv:2303.11366 (2023).

---

> > > ### Author Response · Authors · 2023-11-21
> > > **Request to Reviewer PXAF**
> > >
> > > Reviewer PXAF,
> > >
> > > We have made our best efforts to address your concerns. If they appear satisfactory, we kindly request that you take a moment to revisit your scores and share your updated thoughts. Please also inform us if you have any follow-up questions, and we will do our best to provide clarification before the rebuttal period concludes. Thank you!

---

> > > ### Comment · Reviewer_PXAF · 2023-11-22
> > >
> > > While I acknowledge that self-refine/self-debug/self-reflection could help the model to recover from the errors, there is typically some feedback provided such as execution traces. I am not convinced that the same model can reason about its own error without high-quality feedbacks. The authors mentioned expert review, which can serve as a guard, but doing so on each example level is simply impossible. I still have a lot of concerns on the reliability of the error analysis conducted in this paper through model explanations.

---

> > > > ### Author Response · Authors · 2023-11-22
> > > > **Response to additional comments from Reviewer PXAF regarding error analysis**
> > > >
> > > > Reviewer PXAF, thank you for taking the time to read through our rebuttal and for the additional comment on the error analysis.
> > > >
> > > > We appreciate your concerns regarding the reliability of our error analysis. We acknowledge that expert review of each example at scale is challenging. Nonetheless, we utilize a representative sample that provides a comprehensive view of common error patterns.
> > > >
> > > > To clarify, in our experiments, we analyzed a significant yet manageable subset of errors across all 16 settings for LLMs, incorporating 2 shot paradigms, 4 models, and 2 prompts for each. We did not include the BERT-based models in this analysis due to their unexplainability. In our context, a ‘mistaken sample’ refers to an independent problem. Specifically for the Causality task, we considered all mistaken samples whenever any setting resulted in an incorrect prediction. This resulted in a total of 14 samples for analysis. For the remaining nine tasks, we analyzed a mistaken sample only if at least three settings incorrectly predicted the same problem. The exact breakdown of mistaken samples for each task is as follows: Ordering (313), Frequency (252), Duration (365), Typical Time (146), Ambiguity Resolution (345), Arithmetic (552), Relation (179), Temporal NLI (218), and Storytelling (77). We conducted an individual analysis for every model that made errors on each question.
> > > >
> > > > **We hope that detailing the exact number of mistaken samples for each task clarifies our methodical and manageable approach to error analysis, which is supplemented by expert review**. We are happy to provide further clarifications as needed.
> > > >
> > > > Thank you!

---

> > ### Comment · Reviewer_PXAF · 2023-11-22
> >
> > Thank you authors for the response. I am still not convinced that the tasks are hard enough to be useful for benchmarking future model developments. Therefore, I will still keep my original score.
> >
> > Regarding the task difficulty, I am not convinced by the argument that some subtasks are difficult and hence the whole benchmark has enough headroom for measuring progress. One can always pick the tail distribution and get low performance from the models. The numbers in Table 2 for 8 out of 10 tasks are close to or higher than 90%, leaving little headroom for improvements. For instance, Caus. has accuracy over 99% for most GPT, Llama2 and PaLM2 models. What is the point of using it to evaluate future models?
> >
> > Regarding the point of upon perturbing the dataset, the model performance drops. I don't think that is just an artificial way to make the tasks look harder for the model.

---

> > > ### Author Response · Authors · 2023-11-22
> > > **Response to additional comments from Reviewer PXAF regarding task difficulty**
> > >
> > > Reviewer PXAF, thank you for taking the time to read through our rebuttal and for the additional comment on task difficulty.
> > >
> > > We understand your concern regarding the high performance on certain tasks and agree that merely selecting difficult subtasks does not inherently justify the benchmark's utility. However, it is important to clarify that the **TRAM benchmark is predicated not solely on the current difficulty but also on its potential to evolve with advancing model capabilities**. The subtasks showing high performance indeed indicate the models' strengths, but **the benchmark is designed to highlight both strengths and weaknesses across a spectrum of tasks**.
> > >
> > > Concerning the tasks where models achieve over 90% accuracy, it is worthwhile to consider that reaching or even surpassing human-level performance is the ultimate goal. In this context, even small increments are significant. The high-performing subtasks serve as a baseline for expected model performance, and the lower-performing subtasks represent areas for potential growth.
> > >
> > > The use of perturbations is not intended to artificially inflate task difficulty, but rather to simulate variance in real-world scenarios that models must be adept at handling. **The ability to maintain high performance in the face of nuanced changes is crucial and reflective of a model's robustness and adaptability**.
> > >
> > > To further address your concerns, **we commit to continuing the expansion of the benchmark with more nuanced and complex examples** that represent the tail ends of the performance distribution, ensuring that even high-performing models are sufficiently challenged. In the temporal reasoning domain, in particular, we aim to establish a standard across a variety of tasks, unlike existing benchmarks that propose tasks one by one without a standard for comparison. We acknowledge the current limitations of the tasks and will certainly continue to refine the problem sets in our benchmark in tandem with the models they are designed to evaluate.
> > >
> > > We hope this additional explanation addresses your concerns and would be happy to provide further clarifications.
> > >
> > > Thank you!

---

### Official Review · Reviewer_Z4nr · 2023-11-01

**Soundness:** 4 excellent
**Presentation:** 4 excellent
**Contribution:** 3 good
**Rating:** 6
**Confidence:** 4

**Summary:**

This paper presents a new LLM-focused benchmark suite named TRAM. It consists of a variety of temporal reasoning tasks, and is aimed at driving progress. Experimental results demonstrate a gap between human-level and machine-level performance, suggesting progress to be had.

**Strengths:**

+ I generally like good benchmarks. I think they serve an important purpose in the community, which is to systematize comparisons (and ideally, drive progress).

+ This benchmark seems well thought out, with a thoughtful and diverse set of temporal reasoning tasks.

+ The empirical results suggest that (1) the benchmark discriminates between different models, highlighting their performance discrepancies, and (2) shows a gap between human and machine performance.

**Weaknesses:**

- I like good benchmarks to be hard. I'm a bit concerned that SOTA performance on this benchmark starts at 84%; this perhaps suggests that the benchmark isn't hard enough.

- I'm a bit concerned that some of the questions were sourced from other benchmarks. This could be problematic if it were, for example, included in a larger suite of benchmarks such as Google's BigBench.  I worry that some questions would be double-counted, leading to incorrect conclusions about model performance.

**Questions:**

What percentage of questions, exactly, come from other benchmarks?

---

> ### Author Response · Authors · 2023-11-21
> **Response to Reviewer Z4nr (Part 1)**
>
> Reviewer Z4nr, thank you for providing the insightful suggestions. We hope the following responses address your concerns.
>
> **Weakness 1: Difficulty of the benchmark.**
>
> We appreciate your concern about the difficulty level of our TRAM benchmark. The purpose of this benchmark is to provide a comprehensive test that covers a wide range of temporal reasoning abilities for LLMs. As detailed in our paper, most tasks consist of multiple subtasks. Although the average state-of-the-art (SOTA) performance is high, particularly for 8 out of 10 tasks with accuracies close to or exceeding 90%, there are 12 out of 35 subtasks where the GPT-4 (5S, CoT) setting performs at or below 80%. For instance, in this setting, the Ambiguity Resolution (calendar-shift) task achieves 53% accuracy, the Typical Time (comparison) task 77%, and the Arithmetic (date computation) task 78%. These results underscore specific areas that require further refinement, even in SOTA models.
>
> Meanwhile, to address your concern and challenge the models more rigorously, we have considered some straightforward and effective methods, including introducing additional options and modifying question formats. To test the efficacy of these methods, we conducted case studies on the Causality and Storytelling tasks, which initially showed high performance. Under GPT-4 (5S, CoT), we reevaluated 50 samples from each task that were previously predicted correctly. When the distracting choices were contextually close to the correct answer, the model's performance dropped, with 9 out of 50 incorrect predictions for Causality and 17 out of 50 for Storytelling. In contrast, when distractors were clearly differentiated from the correct answer, the model had 5 out of 50 incorrect predictions for Causality and 13 out of 50 for Storytelling.
>
> Even in a SOTA setting, our deliberate introduction of additional options can decrease model performance by more than 10%. Additionally, converting some tasks to a short-answer format has been shown to affect performance; for example, the SOTA performance on the Arithmetic task dropped from 94.3% to 89.6%. We are dedicated to continuously refining the tasks in our benchmark and to providing a variety of question formats to meet different levels of model testing. Updates to question sets and results will be regularly shared on our GitHub page.
>
> **Weakness 2: Questions from other benchmarks.**
>
> Thank you for your feedback regarding the use of questions from existing datasets in our benchmark. We acknowledge this as a limitation of our work and understand the potential issues it might pose, particularly in the context of larger benchmark suites where double-counting could lead to skewed conclusions about model performance. Similar to the GLUE and SuperGLUE benchmarks, our adoption of pre-existing datasets is based on their implicit acceptance within the NLP community. For questions sourced from existing datasets, we conducted significant preprocessing and reformulation to align them with the specific objectives and format of TRAM. Our preprocessing involved adapting the questions to effectively test temporal reasoning capabilities and reformatting them to fit the structure of our benchmark. This process included manually adding distracting or confusing options, filtering out irrelevant questions for quality control, and reformulating problems. Due to this extensive manual preprocessing, the overlap with larger benchmark suites like Google’s BigBench should be minimal. To further address your concern and enhance LLM evaluation, we have undertaken the following actions and plans:
>
> 1. We have conducted some case studies on tasks in our benchmark, including Causality and Storytelling, by a) adding additional options and b) modifying question formats. The reason for experimenting with a) is to challenge models, potentially confuse them, and test the nuanced reasoning capabilities of LLMs. On the other hand, b) is implemented to avoid the possibility that models can guess instead of truly reasoning. These direct methods have proven effective in our case studies, as demonstrated by the decrease in model performance. Meanwhile, even if models like GPT-4 have been exposed to existing datasets during pretraining, we can still present additional challenges, such as confusing options and challenging problem formats, to test their abilities.

---

> > ### Author Response · Authors · 2023-11-21
> > **Response to Reviewer Z4nr (Part 2)**
> >
> > **Weakness 2: Questions from other benchmarks.**
> >
> > 2. We will prepare and craft more challenging questions for our benchmark, such as reframing existing tasks by introducing novel contexts or counterfactual elements, and introducing a set of novel tasks that require non-linear temporal reasoning. These tasks might include analyzing events in reverse chronological order or understanding cyclical time concepts, which are less likely to be present in standard training datasets. We are dedicated to continuously refining the tasks in our benchmark and providing a variety of question formats to meet different levels of model testing. Updates to question sets and results will be regularly shared on our GitHub page.
> >
> > **Question 1: Percentage of questions from other benchmarks.**
> >
> > Regarding the proportion of questions in our TRAM benchmark that are sourced from other benchmarks, here is a precise breakdown by task. It is presented as the number of sourced questions over the total number of questions, along with the corresponding percentages.
> >
> > | Task                | Percentage                  |
> > |---------------------|-----------------------------|
> > | Ordering            | 1,995 / 29,462 ≈ 6.8%       |
> > | Frequency           | 510 / 4,658 ≈ 10.9%         |
> > | Duration            | 1,297 / 7,232 ≈ 17.9%       |
> > | Typical time        | 9,310 / 13,018 ≈ 71.5%      |
> > | Ambiguity Resolution | 0 / 3,649 = 0%              |
> > | Arithmetic          | 0 / 15,629 = 0%             |
> > | Relation            | 102,462 / 102,462 = 100%    |
> > | Temporal NLI        | 282,144 / 282,144 = 100%    |
> > | Causality           | 31 / 1,200 ≈ 2.6%           |
> > | Storytelling        | 67,214 / 67,214 = 100%      |
> >
> > For problems sourced from existing benchmarks, we conducted significant preprocessing and reformulation to align these questions with the specific objectives and format of TRAM. Our preprocessing involved adapting the questions to effectively test temporal reasoning capabilities and reformatting them to fit the structure of our benchmark. This process included manually adding distracting or confusing options, filtering out irrelevant questions for quality control, and reformulating problems. Similar to the GLUE and SuperGLUE benchmarks, our use of pre-existing datasets is based on their implicit acceptance within the NLP community. Although we built upon existing datasets to varying extents in 8 out of the 10 tasks, our manual work introduced new dimensions and challenges to comprehensively assess the temporal reasoning capabilities of LLMs.

---

### Official Review · Reviewer_v5US · 2023-11-05

**Soundness:** 3 good
**Presentation:** 3 good
**Contribution:** 3 good
**Rating:** 8
**Confidence:** 4

**Summary:**

This paper proposes  a benchmarking dataset for temporal reasoning named TRAM. The TRAM benchmark evaluation is exemplified with BERT-style pretrained-finetuned models  and GPT style prompting-based LLMs. The authors intended to provide TRAM as a comprehensive benchmark to spur the LLM research progress in temporal reasoning capabilities.

**Strengths:**

• A comprehensive benchmark covers various temporal reasoning abilities: ordering, frequency, duration, typical time, ambiguity, arithmetic, relation, temporal NLI, causality, storytelling.
• The overall  size of the dataset is big, being 526,068 problems for benchmarking.
• Pretraining-finetuning and prompting paradigms of LLMs are both evaluated using the benchmarking providing reasoning evaluation conclusions. It is a good starting point from which the community can evolve the LLM techniques  or other LLM alternatives for temporal reasoning.

**Weaknesses:**

• The benchmark currently is only in the form of multi-choice questions.
•  The sizes of different categories of problems are imbalanced. For example, causality is of only 600 problems. This might render the benchmarking evaluation results misguiding. Especially for the pretraining-finetuning paradigms.
•  The texts are mostly from existing datasets. Latest LLMs might have seen them through the pretraining phrase crawled dataset. It might make the benchmarking results over-estimate the performance of LLMs in the temporal abilities. It is an issue beyond just the temporal reasoning abilities extending to all other LLM benchmarking datasets. It calls for more organic benchmarking approaches for LLMs and their iteration which can be pretrained with all kind of available data in human world including benchmarking data.

**Questions:**

1. For the terms "commonsense", "analogy inference", "comparison" and so on, it would be better to have a formal definition and ensure that the datasets and the reasoning follow the formal definitions with verifiable criteria (automatically verifiable would be even better).
2. Page 7, it would be better to include a more comprehensive list of the settings for human expert annotators evaluation. For example, how the experts are drawn from the population, how to ensure they are capable experts or their level of expertise. How will an ordinary person performance comparing to experts? Such a systematic study of human experiments might also provides hints for comparing human performance variation with LLM performance variation.
3. Page 18, appendix B, for self-contained please detail a little bit more for SP with examples. The whole section is essentially about CoT with little information regarding how SP is constructed.

---

> ### Author Response · Authors · 2023-11-21
> **Response to Reviewer v5US (Part 1)**
>
> Reviewer v5US, thank you for providing the constructive suggestions. We hope the following responses address your concerns.
>
> **Weakness 1: Question format.**
>
> We employ multiple-choice questions (MCQs) in TRAM to align with established practices in LLM benchmarks, such as the AI2 Reasoning Challenge (ARC) and Massive Multitask Language Understanding (MMLU), which also utilize MCQs. Additionally, the MCQ format is the most straightforward method for evaluating LLMs. Nevertheless, we recognize the potential limitations of this format. On our GitHub page, we have provided an alternative version of the benchmark in a short-answer format where appropriate, except for tasks like storytelling, ambiguity resolution, and causality. For these three types of questions, I have retained the MCQ format, as other formats could lead to answers that are too open-ended and difficult to evaluate. As part of our ongoing research, we remain open to incorporating more diverse question formats and adding more challenging questions to each task. These updates will be made available on our GitHub page, adapting to the evolving needs of the research community and the developing capabilities of LLMs.
>
> **Weakness 2: Imbalanced problems per category.**
>
> Our causality task is inspired by the Choice of Plausible Alternatives (COPA) dataset, which itself comprises only 1,000 questions. Given that COPA has set a precedent in the field, particularly for tasks centered around causal reasoning, the number of questions it contains serves as a benchmark for what can be considered sufficient for a robust evaluation. Furthermore, 93.5% of the questions in our original causality task are meticulously crafted by hand. This manual process, while ensuring the precision and applicability of each question, also limits the feasible scale of our dataset. To address the limited size of the causality category, we have added an additional, mirrored instance for each original instance. This expands our causality task to a total of 1,200 questions. Each mirrored instance uses the same alternatives as the corresponding original instance but introduces a new premise that matches the incorrect alternative of the original instance. We have updated the relevant information on this specific task in our revised paper, mainly including updated BERT-based results for causality and the task description. We are committed to continuously refining and potentially expanding this task, with updates to the question set being regularly posted on our GitHub page. This effort is specifically aimed at mitigating issues associated with the pretraining-finetuning paradigms, particularly those arising from limited training samples.
>
> **Weakness 3: Texts based on existing datasets.**
>
> Thank you for your feedback regarding the use of questions from existing datasets in our benchmark. We acknowledge this as a limitation of our work and understand the issues it might pose, particularly in terms of potential overestimation of model capabilities. Similar to the GLUE and SuperGLUE benchmarks, our adoption of pre-existing datasets is based on their implicit acceptance within the NLP community. For questions sourced from existing datasets, we conducted significant preprocessing and reformulation to align them with the specific objectives and format of TRAM. Our preprocessing involved adapting the questions to effectively test temporal reasoning capabilities and reformatting them to fit the structure of our benchmark. This process included manually adding distracting or confusing options, filtering out irrelevant questions for quality control, and reformulating problems. Due to this extensive manual preprocessing, the overlap with larger benchmark suites like Google’s BigBench should be minimal. To further address your concern and enhance LLM evaluation, we have undertaken the following actions and plans:
>
> 1. We have conducted some case studies on tasks in our benchmark, including Causality and Storytelling, by a) adding additional options and b) modifying question formats. The reason for experimenting with a) is to challenge models, potentially confuse them, and test the nuanced reasoning capabilities of LLMs. On the other hand, b) is implemented to avoid the possibility that models can guess instead of truly reasoning. These direct methods have proven effective in our case studies, as demonstrated by the decrease in model performance. Meanwhile, even if models like GPT-4 have been exposed to existing datasets during pretraining, we can still present additional challenges, such as confusing options and challenging problem formats, to test their abilities.

---

> > ### Author Response · Authors · 2023-11-21
> > **Response to Reviewer v5US (Part 2)**
> >
> > **Weakness 3: Texts based on existing datasets.**
> >
> > 2. We will prepare and craft more challenging questions for our benchmark, such as reframing existing tasks by introducing novel contexts or counterfactual elements, and introducing a set of novel tasks that require non-linear temporal reasoning. These tasks might include analyzing events in reverse chronological order or understanding cyclical time concepts, which are less likely to be present in standard training datasets. We are dedicated to continuously refining the tasks in our benchmark and providing a variety of question formats to meet different levels of model testing. Updates to question sets and results will be regularly shared on our GitHub page.
> >
> > **Question 1: Task formal definition.**
> >
> > Following your suggestion, we have added a glossary with definitions of all tasks and subtasks in our benchmark to the revised paper (Appendix E, pages 22-23).
> >
> > **Question 2: Human expert details.**
> >
> > Following your suggestion, we have added more details about human experts in our revised paper (Appendix A, page 12).
> >
> > **Selection of Expert Annotators**: Our selection criteria for expert annotators emphasized expertise in temporal reasoning as well as quantitative analysis. This necessitated including professionals with advanced degrees (M.S. or Ph.D.) in cognitive science and psychology, for their insights into the nuances of human temporal cognition, and in statistics, mathematics, and computer science, for their ability to handle the analytical rigor required by many tasks in our benchmark.
> >
> > **Expertise Verification**: We conducted a comprehensive screening process to ensure high levels of expertise. This process involved validating educational qualifications, reviewing related professional or research experience in time perception and quantitative problem-solving, and administering a preliminary assessment involving temporal reasoning problems. One random problem per subtask (38 problems in total) was used, and the criterion for passing the screening was an average accuracy of more than 92% (a maximum of three incorrect problems).
> >
> > **Comparison with Unspecialized Individuals**: We also assessed multiple human non-specialists using our benchmark sourced from Amazon Mechanical Turk. They responded to the same set of 1,900 questions as the experts. The accuracy rate of these non-specialists across all tasks was 62.5%.
> >
> > **Question 3: SP examples.**
> >
> > Thank you for pointing out the missing pieces. In accordance with your suggestions, we have added more details about SP with corresponding examples on Page 18 (Appendix B) of our revised manuscript.

---

> > > ### Comment · Reviewer_v5US · 2023-11-23
> > >
> > > Thank the authors for clarifying my questions. Please improve the paper as other reviewers suggested. I don't have further questions. I will keep my original rating.

---

### Official Review · Reviewer_rYw6 · 2023-11-05

**Soundness:** 2 fair
**Presentation:** 4 excellent
**Contribution:** 3 good
**Rating:** 6
**Confidence:** 3

**Summary:**

This paper introduces a temporal reasoning benchmark for large language models. The benchmark is run on a collection of ten datasets containing thirty-eight subtasks related to time reasoning problems. The datasets used for the temporal reasoning task are formulated as multiple-choice problems, reconstructed from several existing datasets. The paper evaluates the performance of several LLMs on the curated datasets in few-shot learning settings. The experimental results show that there is still room for improvement in enhancing the temporal reasoning abilities of these models.

**Strengths:**

1, The author introduces a new dataset and benchmark for evaluating the temporal reasoning capabilities of large language models with sufficient amounts of data in different time-related domains, including duration, frequency, ordering, etc.

2, The author provides an in-detail description of the format of the benchmark dataset.

3, The author provides a comprehensive experimental evaluation of popular LLMs, including GPT-4, GP3-3.5, and Llamma2 on the TRAM benchmark.

4, the author provides error analysis on different task groups, this can help researchers prioritize their efforts and further improve the temporal reasoning abilities of LLMs in the future.

**Weaknesses:**

1, As the paper primarily focuses on the area of datasets and benchmarks in large language models, it is better to provide an anonymous GitHub page, for example (https://anonymous.4open.science/) with code for dataset curation and empirical evaluation, as well as simple documentation on running the LLM’s assessments.

---

2, At this point, the overall contribution of the dataset curation done by the authors is unclear. It is better to provide some examples for comparing the differences between the source dataset and the provided curated dataset.
With some manual comparison between the shared supplementary materials and the source dataset repos (https://github.com/CogComp/MCTACO/tree/master) and (https://rajpurkar.github.io/SQuAD-explorer/). The author seems to simply reformulate multiple original Yes/No questions in one multiple-choice question. For example, in the ‘frequency’ task, the original question from MCTACO looks like:

Q1: For example, buy local produce at a farmers market, like the one in Figure 2.20. How often do they buy produce from the farmers market?
twice a week	yes	Frequency

Q2: For example, buy local produce at a farmers market, like the one in Figure 2.20. How often do they buy produce from the farmers market?	he says for example	no	Frequency

……

Q8: For example, buy local produce at a farmers market, like the one in Figure 2.20. How often do they buy produce from the farmers market?	once a second	no	Frequency

However, the provided dataset in this paper takes the above 8 original questions and reformulates the question as (in file frequency.csv):

Q1: For example, buy local produce at a farmers market, like the one in Figure 2.20. How often do they buy produce from the farmers market?

Answer A: twice a second

Answer B: he says for example

Answer C: twice a week

Correct Answer: C

First, the reconstructed candidate choices contain answers that do not correlate with time (Answer B), which is caused by the error from the original data source. It’s better to provide an overall evaluation of the data quality.  Second, it is better to explain the advantages of converting several existing Yes/No questions to one multiple-choice question.

---

3, It is better to provide an overall description of the dataset following the datasheet for datasets [1] (or other similar sources), I believe this may address most of the concerns.

---

4, The experimental results show a disparity in performance across different Large Language Models (LLMs). In addition, with the integration of chain-of-thought prompting, the results show only minor improvements. In addition, GPT-4 appears to outperform every other model by a large margin. It is unclear whether the provided model is already trained on the source dataset (MCTACO) from which this benchmark dataset is derived. Here is the answer I got from the GPT-3.5 and GPT-4 by prompting the question: ‘Please provide a detailed description of the MCTACO dataset for temporal reasoning.’

GPT-3.5:
I'm sorry, but as of my last knowledge update in January 2022, I do not have specific information about the "MCTACO" dataset for temporal reasoning. It's possible that this dataset was created or became publicly available after my last update, or it may not be a well-known dataset in the field of natural language processing……..

GPT-4:
The MCTACO (Multiple Choice Temporal Commonsense Reasoning Assessment) dataset is a collection of questions designed for evaluating the temporal commonsense reasoning abilities of machine learning models. This dataset is particularly focused on the aspect of understanding time-based common sense or temporal common sense, which is essential for natural language understanding systems…….
It is better to provide some justifications for the problem mentioned above.

---

[1]: Gebru, Timnit, et al. "Datasheets for datasets." Communications of the ACM 64.12 (2021): 86-92.

**Questions:**

My questions are listed in the weakness, and can be summarized into three folds:

1, Did the author do a lot of data preprocessing on curating the benchmark dataset?

2, What are the advantages of reformulating the original data sources into the TRAM dataset? What is the major differences between the source dataset and the curated dataset?

3, What causes the significant disparity in model performance? Is it because some LLMs are already trained on the source dataset?

---

> ### Author Response · Authors · 2023-11-21
> **Response to Reviewer rYw6 (Part 1)**
>
> Reviewer rYw6, thank you for providing the detailed suggestions. We hope the following responses address your concerns.
>
> **Weakness 1: GitHub page.**
>
> Following your suggestion, we have set up an anonymous GitHub page. Here is the link: https://anonymous.4open.science/r/TRAM-Benchmark-596D/.
>
> **Weakness 2/Question 2: Overall contribution of the work.**
>
>  Thank you for your insightful suggestion about clarifying the contribution of our work and the rationale behind the transformation of existing data sources into the TRAM dataset. The following table provides a comparison between our TRAM and previous temporal reasoning datasets, including task coverage and data size. The order of the prior work listed is from the most recent to the earlier ones.
>
> | Dataset/Benchmark | Size  | Task Coverage                                                  |
> |-------------------|-------|---------------------------------------------------------------|
> | TRAM              | 526.7k| Ordering, Frequency, Duration, Typical Time, Ambiguity Resolution, Arithmetic, Relation, Temporal NLI, Causality, Storytelling |
> | MenatQA           | 2.9k  | Scope, Order, Counterfactual                                   |
> | TEMPREASON        | 52.8k | Facts                                                          |
> | TEMPLAMA          | 50k   | Facts                                                          |
> | Time-Sensitive QA | 41.2k | Facts                                                          |
> | TIMEDIAL          | 1.1k  | Commonsense                                                    |
> | MCTACO            | 13k   | Commonsense (duration, temporal ordering, typical time, frequency, stationarity) |
>
> Compared to earlier works, our dataset offers a much larger data size and broader task coverage, encompassing 10 main tasks and 38 subtasks. For instance, in the classic tasks of ‘Duration’ and ‘Frequency’, earlier works mainly cover Facts and Commonsense. In contrast, our work includes 7 subtasks for ‘Duration’ and 6 subtasks for ‘Frequency’, delving into more nuanced aspects of these temporal elements, which are less emphasized by earlier works. Consequently, the extensive data size of TRAM and the intricate design of its subtasks represent a notable change in both the depth and breadth of temporal reasoning (TeR) evaluation, compared to existing benchmarks.
>
> Following your suggestion, we have included examples in Appendix F (pages 23-24) that compare our curated dataset with the source datasets, specifically MCTACO and SQuAD. For information on data quality, please refer to the datasheets available on our anonymous GitHub page. The reformulation of original data sources into the TRAM benchmark offers several advantages:
>
> **Increased Complexity and Depth**: By converting simple Yes/No questions to multiple-choice format, we inherently increase the complexity of each question. This change demands that models not only identify the correct answer but also discern and reject multiple plausible but incorrect options, thereby enhancing the depth of temporal reasoning evaluation.
>
> **Reduction in Guesswork**: The multiple-choice format reduces the likelihood of correct guesses compared to Yes/No questions, demanding a more nuanced understanding and reasoning, and thus providing a clearer assessment of a model's temporal reasoning capabilities.
>
> In summary, we have chosen to use existing benchmarks due to their wide acceptance within the NLP community. The multiple-choice format, effective for evaluating LLMs as seen in benchmarks like the AI2 Reasoning Challenge (ARC) and Massive Multitask Language Understanding (MMLU), introduces more complexity for the model and challenges it to discern subtle differences in temporal reasoning contexts. Our benchmark combines the strengths of existing benchmarks with extensive manual work, setting a new standard for assessing temporal reasoning in LLMs. To address the evolving needs of advanced model evaluation, we will regularly share updates to our question sets and results on our GitHub page.
>
> **Weakness 3/Question 1: Overall dataset description.**
>
> Thank you for providing the insightful advice. Following your suggestions, we have created a datasheet for the TRAM benchmark, which is available on the anonymous GitHub page (datasheet_for_TRAM_benchmark.pdf). Please let us know if there are any additional aspects you think would be worthwhile to incorporate into the datasheet. We are open to suggestions and will gladly update it accordingly.

---

> ### Author Response · Authors · 2023-11-21
> **Response to Reviewer rYw6 (Part 2)**
>
> **Weakness 4/Question 3: Performance disparity among LLMs.**
>
> Thank you for your feedback regarding the use of questions from existing datasets in our benchmark. We acknowledge this as a limitation of our work and understand the issues it might pose, particularly in terms of performance disparity among different LLMs. Similar to the GLUE and SuperGLUE benchmarks, our adoption of pre-existing datasets is based on their implicit acceptance within the NLP community. For questions sourced from existing datasets, we conducted significant preprocessing and reformulation to align them with the specific objectives and format of TRAM. Our preprocessing involved adapting the questions to effectively test temporal reasoning capabilities and reformatting them to fit the structure of our benchmark. This process included manually adding distracting or confusing options, filtering out irrelevant questions for quality control, and reformulating problems. The performance disparity, especially the superior results of GPT-4, can be attributed to varying architectures, training dataset scales, and sizes of the models. While GPT-4 may have been exposed to these datasets during pretraining, our manual work in reformulating the tasks aims to introduce new dimensions and challenges, thus minimizing any potential advantage. To address both this issue and the need for advanced LLM evaluation, we have undertaken the following actions and plans:
>
> 1.  We have conducted some case studies on tasks in our benchmark, including Causality and Storytelling, by a) adding additional options and b) modifying question formats. The reason for experimenting with a) is to challenge models, potentially confuse them, and test the nuanced reasoning capabilities of LLMs. On the other hand, b) is implemented to avoid the possibility that models can guess instead of truly reasoning. These direct methods have proven effective in our case studies, as demonstrated by the decrease in model performance. Meanwhile, even if models like GPT-4 have been exposed to existing datasets during pretraining, we can still present additional challenges, such as confusing options and challenging problem formats, to test their abilities.
>
> 2. We will prepare and craft more challenging questions for our benchmark, such as reframing existing tasks by introducing novel contexts or counterfactual elements, and introducing a set of novel tasks that require non-linear temporal reasoning. These tasks might include analyzing events in reverse chronological order or understanding cyclical time concepts, which are less likely to be present in standard training datasets. We are dedicated to continuously refining the tasks in our benchmark and providing a variety of question formats to meet different levels of model testing. Updates to question sets and results will be regularly shared on our GitHub page.

---

> > ### Author Response · Authors · 2023-11-21
> > **Request to Reviewer rYw6**
> >
> > Reviewer rYw6,
> >
> > We have made our best efforts to address your concerns. If they appear satisfactory, we kindly request that you take a moment to revisit your scores and share your updated thoughts. Please also inform us if you have any follow-up questions, and we will do our best to provide clarification before the rebuttal period concludes. Thank you!

---

> > > ### Comment · Reviewer_rYw6 · 2023-11-22
> > > **Response to the author**
> > >
> > > Thank you so much for the detailed clarification and thank you again for providing an anonymous GitHub repository and datasheet for the TRAM benchmark.
> > >
> > > ---
> > > However, I still have the following concerns:
> > >
> > > I ran several data generation codes, and I noticed that for some topics, a proportion of the questions are generated in a simple format.
> > >
> > > For example, In Arithmetic tasks (https://anonymous.4open.science/r/TRAM-Benchmark-596D/data_processing/arithmetic.ipynb)
> > > I have the following output questions from ‘df_month’:
> > >
> > > ---
> > >
> > > Question 1: Which month comes 3 months after April?
> > >
> > > Question 2: Which month was 4 months before December?
> > >
> > > ……
> > >
> > > Question 500: Which month was 2 months before February?
> > >
> > > And ‘df_time_op’ gives questions like:
> > >
> > > Question 1: Convert 214 minutes into hours.
> > >
> > > Question 2: Convert 398 minutes into hours.
> > >
> > > ……
> > >
> > > Question 1500: Convert 4 days into minutes.
> > >
> > > ---
> > >
> > > In addition, some questions are generated with limited templates, for example in frequency tasks (https://anonymous.4open.science/r/TRAM-Benchmark-596D/data_processing/frequency.ipynb):
> > >
> > > I have the following outputs from ‘df_abstract_frequency’:
> > >
> > > ---
> > >
> > > Question 1: If an athlete trains every 2 days, starting on Wednesday, on which day will they train next?
> > >
> > > Question 2: A town hosts a carnival every 4 years in November. If the preceding carnival was in 1937, when will the subsequent one occur?
> > >
> > > Question 3: On Planet Alpha, 1 day is equivalent to 80 Earth days. How many Earth days elapse between daily events on Alpha?
> > >
> > > ……
> > >
> > > Question 997: A town hosts a carnival every 2 years in September. If the preceding carnival was in 1895, when will the subsequent one occur?
> > >
> > > Question 998: A town hosts a carnival every 3 years in February. If the preceding carnival was in 1942, when will the subsequent one occur?
> > >
> > > Question 999: In a spaceship experiencing time dilation, 1 year inside equates to 37 Earth years. If an event occurs inside every year, how frequently is it observed from Earth?
> > >
> > > Question 1000: A town hosts a carnival every 5 years in January. If the preceding carnival was in 1845, when will the subsequent one occur?
> > >
> > > ---
> > >
> > > Where these 1000 questions are generated using 10 templates. Similarly, the Ambiguity resolution tasks: (https://anonymous.4open.science/r/TRAM-Benchmark-596D/data_processing/ambiguity_resolution.ipynb), where 300 questions are constructed by converting calendars between Gregorian, Julian, Hebrew, and Islamic:
> > >
> > > Question 1: If the date is 9/6/1868 in the Gregorian, what is the date in the Hebrew?
> > >
> > > Question 2: If the date is 5/7/1805 in the Gregorian, what is the date in the Islamic?
> > >
> > > ……
> > >
> > > Question 300: If the date is 3/23/1814 in the Gregorian, what is the date in the Julian?
> > >
> > > ---
> > >
> > > **However, I do appreciate the efforts in some other subtask datasets like:**
> > >
> > > 'df_facts', 'df_multistep_comparison' and 'df_inference' from duration (https://anonymous.4open.science/r/TRAM-Benchmark-596D/data_processing/duration.ipynb)
> > >
> > > 'df_implicit_phrases' from ambiguity resolution (https://anonymous.4open.science/r/TRAM-Benchmark-596D/data_processing/ambiguity_resolution.ipynb)
> > >
> > > ---
> > >
> > > Also, according to the provided source code, I believe the author has made enough effort to curate this dataset. Therefore I adjusted my score from 5 to 6 to recommend borderline acceptance.

---

> > > > ### Author Response · Authors · 2023-11-22
> > > > **Response to additional comments from Reviewer rYw6**
> > > >
> > > > Reviewer rYw6,
> > > >
> > > > We are deeply grateful for your acknowledgment of our efforts and for your re-evaluation of our work, which has led to an improved acceptance score. Thank you for engaging with the TRAM benchmark with such depth.
> > > >
> > > > We appreciate your bringing to our attention the concerns regarding the simplicity of some generated questions and the limited templates used in some tasks. In response to your feedback, we will ensure to:
> > > >
> > > > 1. Increase the variety of questions by expanding the template base as necessary, thus ensuring a broader range of complexity within each task.
> > > > 2. Introduce a higher proportion of complex question types, particularly for those currently in high SOTA performance.
> > > > 3. Separate the problem sets of each (sub)task into different difficulty levels for flexible model evaluation.
> > > > 4. Provide regular updates to the benchmark on the GitHub page and continuously invite further community feedback to refine the dataset.
> > > >
> > > > Your input has been invaluable, and we remain open to further suggestions and discussions. Thank you again!

---

### Official Review · Reviewer_fFct · 2023-11-06

**Soundness:** 3 good
**Presentation:** 4 excellent
**Contribution:** 3 good
**Rating:** 6
**Confidence:** 4

**Summary:**

This work introduces various temporal reasoning tasks in MCQ format with a single correct answer. These tasks include Ordering, Frequency, Duration, Time, Ambiguity Resolution, Arithmetic, Relation, Temporal NLI, Causality, Storytelling. This work also includes the results of these datasets on various LLMs like GPT-4, GPT-3.5, Llama2, and Palm2 in settings like zero-shot, few-shot with standard prompting, and Chain of Thought. Furthermore, they also include results from BERT and RoBERTa. The authors also did a thorough error analysis to find out where the models were going wrong and gain a better understanding of the mistakes the models were making.

**Strengths:**

* Detailed description of dataset creation, sources, templates, and prompts used.
* Insightful error analysis, which investigated every specific error type at a task level.
* Results on several LLMS like GPT-4/3.5, Llama2, Palm2

**Weaknesses:**

* There are many specifically designed models to solve temporal reasoning. None of these models are included in the benchmarks. Without these, it is difficult to compare results between LLMs and RoBERTa or BERT. What goodness that LLMs bring in which tasks compared to these special models which are smaller compared to LLMs?

[1] Yuan, Weizhe, and Pengfei Liu. "reStructured Pre-training."  (2022)

[2] Ben Zhou, Kyle Richardson, Qiang Ning, Tushar Khot, Ashish Sabharwal, and Dan Roth. (2021). "Temporal Reasoning on Implicit Events from Distant Supervision."

**Questions:**

* How does the length of questions in MCQ affect performance? And adding more options, will it confuse models?
* In error analysis, how many mistaken samples were analyzed for each task?

---

> ### Author Response · Authors · 2023-11-21
> **Response to Reviewer fFct (Part 1)**
>
> Reviewer fFct, thank you for providing the insightful suggestions. We hope the following responses address your concerns.
>
> **Weakness 1: Temporal reasoning model comparison.**
>
> Thank you for highlighting the need for a comparison with domain-specific models. Following your suggestion, we have conducted additional experiments using a temporal reasoning model (https://huggingface.co/GAIR/rst-temporal-reasoning-11b), referred to as RST-TeR below, where the training and testing sets are the same as those used for BERT-based models. We have also included results from RoBERTa-large and GPT-4 for reference and comparison:
>
> | Model            | Order | Freq. | Dur. | Typical Time | Amb. Res. | Arith. | Rel.       | NLI        | Caus.      | Story | Average |
> |-------------------|-------|------|------|--------------|-----------|--------|------------|------------|------------|-------|---------|
> | GPT-4 (5S, CoT)  | 71.5        | 93.7 | 93.0         | 90.2      | 89.8   | 94.3       | 69.5/69.1  | 90.7/91.0  | 100.0 | 96.3    | 87.4       |
> | RoBERTa-large    | 55.5        | 57.7 | 55.4         | 60.0      | 41.0   | 29.1       | 90.0/90.0  | 70.0/70.3  | 88.0  | 84.0    | 65.9       |
> | RST-TeR          | 54.5        | 56.2 | 52.3         | 58.7      | 39.8   | 31.6       | 91.5/91.6  | 68.2/68.7  | 87.5  | 82.2    | 65.2       |
>
> Based on these results, RST-TeR achieves performance on par with the RoBERTa-large model. However, there is a significant gap between this domain-specific model and LLMs like GPT-4, except in the Relation (Rel.) task, where RST-TeR is specifically trained. GPT-4’s extensive training across varied contexts contributes to its robust performance, showcasing its superior generalization capabilities. Overall, LLMs bring depth and versatility to temporal reasoning, enabling them to handle a range of tasks that demand a comprehensive and integrative approach, an aspect where specialized models may be more limited.
>
> **Question 1: Impact of length of questions and number of options on model performance.**
>
> Thank you for raising the insightful question about how the length of questions and the number of options in MCQs affect model performance. Here are the statistics for the average length of questions, options, and their combined average for each task:
>
> | Category             | Avg. Question Length | Avg. Option Length | Avg. Combined Length |
> |----------------------|----------------------|--------------------|----------------------|
> | Ordering           |        56.8                 | 7.8                | 64.6                 |
> | Frequency            | 25.3                 | 9.3                | 34.6                 |
> | Duration             | 41.7                 | 8.0                | 49.7                 |
> | Typical Time         | 98.1                 | 4.6                | 102.7                |
> | Ambiguity Resolution | 20.7                 | 5.2                | 25.9                 |
> | Arithmetic           | 13.3                 | 6.9                | 20.2                 |
> | Relation             | 53.9                 | 3.3                | 57.2                 |
> | NLI                  | 44.1                 | 3.0                | 47.1                 |
> | Causality            | 14.2                 | 16.5               | 30.7                 |
> | Storytelling               | 48.3                 | 17.7               | 66.0                 |
>
> Based on our experiments, we have the following observations:
>
> **Length of questions vs. Performance**: There is no direct correlation between question/option length and model performance. For instance, tasks with longer questions such as Typical Time and Storytelling have some of the highest accuracies, indicating that the additional context from longer questions can be beneficial. Conversely, shorter tasks like Arithmetic and Ambiguity Resolution do not necessarily result in higher performance, suggesting that brevity, which may reduce context, does not automatically improve performance. For the tasks in our study, as there is no substantial difference in length, the intrinsic task complexity appears to have a more pronounced effect on the final performance rather than the question length.
>
> **Length of options vs. Performance**: Longer option lengths in tasks like Causality and Storytelling do not lead to uniform decreases in performance, implying that detailed options may provide valuable context that helps the model deduce the correct answer. However, if the added length introduces ambiguity or irrelevant details, it could confuse the model.

---

> > ### Author Response · Authors · 2023-11-21
> > **Response to Reviewer fFct (Part 2)**
> >
> > **Question 1: Impact of length of questions and number of options on model performance.**
> >
> > **Number of options vs. Performance**: Based on our manual preprocessing procedure for constructing distracting options and the experimental results, we find that adding more options can indeed confuse the model. Considering the complexity differences among various tasks, we additionally conducted case studies on tasks that initially performed well, specifically on Causality and Storytelling tasks that lacked a sufficient number of options. We used GPT-4 to evaluate 50 samples for each task that were originally predicted correctly. We considered two scenarios. First, if the distracting choices are closely related to the correct answer contextually, the model incorrectly predicted 9 out of 50 for Causality and 17 out of 50 for Storytelling. Conversely, if there is a clear distinction between the distractions and the correct answer, the model incorrectly predicted 5 out of 50 for Causality and 13 out of 50 for Storytelling. Thus, we conclude that option similarity has a more significant impact on performance than the mere number of options.
> >
> > **Question 2: Number of mistaken samples in error analysis.**
> >
> > In the TRAM benchmark, which comprises ten tasks, we conducted our error analysis across all 16 settings for LLMs, incorporating 2 shot paradigms, 4 models, and 2 prompts for each. We did not include the BERT-based models in this analysis due to their unexplainability. In our context, a ‘mistaken sample’ refers to an independent problem. Specifically for the Causality task, we considered all mistaken samples whenever any setting resulted in an incorrect prediction. This resulted in a total of 14 samples for analysis. For the remaining nine tasks, we analyzed a mistaken sample only if at least three settings incorrectly predicted the same problem. The exact breakdown of mistaken samples for each task is as follows: Ordering (313), Frequency (252), Duration (365), Typical Time (146), Ambiguity Resolution (345), Arithmetic (552), Relation (179), Temporal NLI (218), and Storytelling (77). For each question, we conducted an individual analysis for every model that made errors.

---

> > > ### Comment · Reviewer_fFct · 2023-11-23
> > >
> > > Thank you for the very detailed comments.
> > >
> > > I have a few questions
> > > * In the above results, Is RST-TeR fine-tuned or prompted in a zero-shot or few-shot way?

---

### Author Response · Authors · 2023-11-21
**General Response**

Dear reviewers,

Thank you for the time and effort you have dedicated to reviewing our paper. We have submitted a response to your concerns, along with a revised version of our manuscript. Changes are annotated in red. We would appreciate your feedback on whether our revisions and clarifications have addressed your concerns, or if further clarification is needed. Thank you again!

---

### Meta-Review · Area_Chair_KRpz · 2023-12-05

**Metareview:**

This work presents a benchmark dataset for temporal reasoning. The dataset includes over 500K samples of various subtasks, such as Ordering, Frequency, Duration, Typical Time, Ambiguity Resolution, Arithmetic, Relation, Temporal NLI, Causality, Storytelling. It also provides performance statistics for several LLMs (e.g. Llama2, PaLM2, GPT3.5, GPT4) in zero- and few-shot setups with standard prompting and Chain-of-thought prompting. In addition, it reports results for finetuned BERT and RoBERTa-large.

The authors and reviewers have also had constructive engagements during the discussion period leading to improvements to the paper: For example, authors have clarified their human evaluation strategy, added more details about the source of the dataset, made the code and data and data sheet available through an anonymous link, and added results for a new model particularly geared toward temporal reasoning (RST-TeR).

While this work has potential and is a step in the right direction, the draft in its current shape may not be ready for publication in ICLR due to the following unresolved concerns:

**Task difficulty:** Overall, the goal of a benchmark is to help with development and improvement of future models. However, this dataset in its current shape is not challenging enough. Several of the presented models commonly used today do very well on 8/10 tasks of this benchmark, with little difference compared to human performance and even perfect accuracy. This leaves little room for improvement which minimizes the utility of this dataset. The authors have used several arguments in response to this concern raised by multiple reviewers, but they are not convincing and do not really address the underlying issue raised here:

- *“There are 12 out of 34 subtasks where the GPT-4 (5S, CoT) setting performs at or below 80%.”*
I am assuming the authors are referring to the sum of column 3 in table 1 which suggests 38 tasks. However, the results reported in Table 2 provide a breakdown for 10 categories, and there is no breakdown of performance metrics reported for the subtasks to verify this claim. Additionally, appendix fig 12 suggests that several of these subtasks only include a small proportion of examples, so may not be representative on their own.

- *“TRAM benchmark is predicated not solely on the current difficulty but also on its potential to evolve with advancing model capabilities.”*
However, we cannot accept papers in ICLR main track based on their future potential. That would be a fit for acceptance criteria of a workshop.

- *“small increments are significant”*
 is not a convincing argument.

**Unreliable and potentially problematic error analysis methodology:**
- *“We prompted the model to explain its decisions, then reviewed these explanations to identify errors, understand the reasons behind them, and categorize them into specific error types.”*
Model’s generated explanations are not a reliable source for error analysis. A more reliable approach would be to design the benchmark questions to particularly test for those hypotheses.

**Justifying TRAM design choices and distinguishing them/positioning them with respect to prior benchmarks:**
The authors have definitely made improvements in addressing these concerns by providing more details about TRAM. However, some of the claims made are not verified, and some arguments seem contradictory in response to some of the raised concerns.
-Authors argue that the MCQ setup *“inherently increases the complexity of each question”* [compared to the benchmarks the data was sourced from]. At the same time, also suggesting a short answer setup as a more complex alternative (which is the same as the SQuAD setup), *“Additionally, converting some tasks to a short-answer format has been shown to affect performance; for example, the SOTA performance on the Arithmetic task dropped from 94.3% to 89.6%.”*
-The choice of “same, but not same” may lead to confusion. Authors argue their choice of “pre-existing datasets is based on their implicit acceptance within the NLP community. “ But they also claim making significant changes to these datasets with reformatting questions and options etc, and there is limited discussion on how to compare performance on these subtasks in TRAM and the benchmark dataset they were sourced from.

In summary, this paper has great potential, but would benefit from another round of revision before publication. I recommend the authors take into account the points summarized above to further improve this work before their next submission.

**Justification For Why Not Higher Score:**

While this work has potential and is a step in the right direction, the draft in its current shape may not be ready for publication in ICLR due to the following unresolved concerns, as discussed in the meta-review:

**Task difficulty:** Overall, the goal of a benchmark is to help with development and improvement of future models. However, this dataset in its current shape is not challenging enough. Several of the presented models commonly used today do very well on 8/10 tasks of this benchmark, with little difference compared to human performance and even perfect accuracy. This leaves little room for improvement which minimizes the utility of this dataset. The authors have used several arguments in response to this concern raised by multiple reviewers, but they are not convincing and do not really address the underlying issue raised here:

- *“There are 12 out of 34 subtasks where the GPT-4 (5S, CoT) setting performs at or below 80%.”*
I am assuming the authors are referring to the sum of column 3 in table 1 which suggests 38 tasks. However, the results reported in Table 2 provide a breakdown for 10 categories, and there is no breakdown of performance metrics reported for the subtasks to verify this claim. Additionally, appendix fig 12 suggests that several of these subtasks only include a small proportion of examples, so may not be representative on their own.

- *“TRAM benchmark is predicated not solely on the current difficulty but also on its potential to evolve with advancing model capabilities.”*
However, we cannot accept papers in ICLR main track based on their future potential. That would be a fit for acceptance criteria of a workshop.

- *“small increments are significant”*
 is not a convincing argument.

**Unreliable and potentially problematic error analysis methodology:**
- *“We prompted the model to explain its decisions, then reviewed these explanations to identify errors, understand the reasons behind them, and categorize them into specific error types.”*
Model’s generated explanations are not a reliable source for error analysis. A more reliable approach would be to design the benchmark questions to particularly test for those hypotheses.

**Justifying TRAM design choices and distinguishing them/positioning them with respect to prior benchmarks:**
The authors have definitely made improvements in addressing these concerns by providing more details about TRAM. However, some of the claims made are not verified, and some arguments seem contradictory in response to some of the raised concerns.
-Authors argue that the MCQ setup *“inherently increases the complexity of each question”* [compared to the benchmarks the data was sourced from]. At the same time, also suggesting a short answer setup as a more complex alternative (which is the same as the SQuAD setup), *“Additionally, converting some tasks to a short-answer format has been shown to affect performance; for example, the SOTA performance on the Arithmetic task dropped from 94.3% to 89.6%.”*
-The choice of “same, but not same” may lead to confusion. Authors argue their choice of “pre-existing datasets is based on their implicit acceptance within the NLP community. “ But they also claim making significant changes to these datasets with reformatting questions and options etc, and there is limited discussion on how to compare performance on these subtasks in TRAM and the benchmark dataset they were sourced from.

In summary, this paper has great potential, but would benefit from another round of revision before publication. I recommend the authors take into account the points summarized above to further improve this work before their next submission.

**Justification For Why Not Lower Score:**

N/A

---

### Decision · Program_Chairs · 2024-01-16

Reject